# Conformity to social norm interventions is not amplified in tighter nations
Jane Acierno [1,9], Elisa Tedaldi [2,9], Joel Ginn[3], Danielle Goldwert[4], Madalina Vlasceanu [5],
Sandra J. Geiger [6,7], Gregg Sparkman[8] & Sara M. Constantino [5] ✉

Social norms have a reliable and oftentimes strong influence on individual attitudes and behaviors across environmental and other domains. This influence has been theorized to differ by cultural tightness—the extent to which people adhere to shared cultural norms. Understanding whether and how cultural context moderates the influence of behavioral interventions is essential for the design of culturally-attuned and adaptable interventions to address collective action problems like climate change. Yet, past research has primarily relied on correlational approaches to demonstrate cross-cultural differences in norm adherence, without experimentally manipulating norm information across different settings, limiting insights into how cultural context shapes conformity to norm interventions. Our study tests the effects of three social norm interventions on the climate-related attitudes and behaviors of 16,089 participants in 42 countries. We find that conformity to social norm interventions is not uniformly amplified in tight cultures, suggesting a nuanced relationship between cultural tightness and normative interventions.

The tendency of people to conform to social norms is well-established across the behavioral and social sciences[1-6] and is important for sustaining cooperation, coordination, and mobilizing collective action in domains such as natural resource use and climate change[4,7,8]. As such, psychologists and other social scientists have pointed to norm interventions—using norm information to influence behavior—as one avenue for promoting widespread social change[8-12], and have proposed guidelines describing how to utilize social norms to address global environmental challenges[7,13].

In recent decades, social norm interventions have gained traction across a variety of domains, from public health to bullying[14-17]. They have also been used to promote a range of climate-friendly behaviors including eco-conscious consumerism[18], energy conservation[19-25], waste reduction[26-28], sustainable food consumption[29], water conservation[30-32], and even sustainable infrastructure design[33]. However, despite the promise of social norm interventions to promote social change and help address societal challenges, there are limitations in the existing literature regarding the scalability, durability, and generalizability of social norm interventions to different contexts[34], in part because much of this work originates in the Global North. Over time, researchers have extended social norm interventions across diverse cultural and geographic settings. These studies find

that providing normative information about others' beliefs and behaviors can be effective across the globe[35,36]. At the same time, there are also studies conducted that find mixed evidence for the effectiveness of social norm interventions across different cultural contexts[37-39].

These mixed effects are perhaps unsurprising given that the importance of social norms in governing the behavior of individuals has also been shown to vary across cultural contexts, rendering the strength of conformity to norm interventions in different countries and communities uncertain. Research on cultural tightness—defined as the extent to which social norms are strictly enforced and deviant behavior socially sanctioned—offers a useful theoretical lens for understanding differences in norm adherence across cultures[40-44]. In the past decade, researchers have documented variation in cultural tightness across countries and communities (e.g.[41,45-52]). This research finds that in tighter cultures such as India or Singapore, individuals learn to follow rules and adhere to an existing social order from childhood[43] and are more likely to report adhering to social norms and expectations[53]. This psychological attunement to social expectations facilitates faster detection of social norms and greater behavioral regulation to ensure alignment with the norm[54,55]. In contrast, individuals in looser cultures, such as the Netherlands or Israel, are more likely to prioritize personal

[1]Department of Psychology, Northeastern University, Boston, MA, USA. [2]Department of Communication and Media Research, University of Zurich, Zurich, Switzerland. [3]Department of Psychology, Grinnell College, Grinnell, IA, USA. [4]Department of Psychology, New York University, New York, NY, USA. [5]Department of Environmental Social Sciences, Doerr School of Sustainability, Stanford University, Stanford, CA, USA. [6]Environmental Psychology, Department of Cognition, Emotion, and Methods in Psychology, Faculty of Psychology, University of Vienna, Vienna, Austria. [7]Andlinger Center for Energy and the Environment, Princeton University, Princeton, NJ, USA. [8]Department of Psychology and Neuroscience, Boston College, Chestnut Hill, MA, USA. [9]These authors contributed equally: Jane Acierno, Elisa Tedaldi. ✉e-mail: sara.constantino@gmail.com

values and accept greater attitudinal and behavioral variability[56]. This difference is due in part to tighter cultures having more formalized mechanisms for detecting and enforcing norm violations[43], as well as stronger self-monitoring[57]. As such, individuals in tight cultures—who are culturally attuned to monitor and respond to normative signals—may be more responsive to social norm interventions as they highlight or provide social norm information[54]. Cultural tightness thus offers a valuable theoretical framework for theorizing about how the effectiveness of social norm interventions might vary across contexts. However, despite its theoretical relevance, experimental tests of this relationship are limited.

Much of the work examining conformity to social norms across cultural contexts is correlational[58] and does not provide a systematic or causal test of the effectiveness of social norm interventions across tight and loose cultures. Nonetheless, these studies demonstrate that conformity to social norms does indeed differ across the tightness-looseness spectrum[59,60], suggesting that cultural tightness may moderate the effectiveness of norm interventions by promoting conformity to information about prevailing social norms. Although direct research on the interaction between social norm interventions and cultural tightness is limited, some studies have explored related constructs like individualism-collectivism. Cross-cultural differences in the prioritization of individual versus collective goals[61] have been shown to moderate responses to norm interventions. These studies tend to find higher levels of norm adherence and greater disapproval of socially deviant behavior among individuals from collectivist cultures as compared to individualistic ones[62–66].

Although cultural tightness and individualism are modestly correlated ($r = 0.44$,[67]; $r = -0.47$[41]), they are distinct constructs. While tightness captures the strength of adherence to, and maintenance of, social norms in society, individualism and collectivism refer to the extent to which people are motivated by individual versus collective goals[50,61]. Reflecting this difference, the two constructs correlate with different national-level factors: collectivism correlates with national wealth while tightness correlates with a nation's history of conflict and efforts to maintain social order[41]. There is thus an important gap in the literature when it comes to understanding whether cultural tightness—a measure that specifically captures the importance of adherence to social norms in a society—moderates the effectiveness of commonly used norm interventions. These findings could have implications for the design of behavioral interventions aimed at addressing societal challenges including in the financial, health, and climate domains. We fill this gap by conducting a large-scale, cross-cultural, experimental analysis of the efficacy of distinct social norm interventions across culturally tight and loose contexts, focusing on the domain of climate change.

This study extends previous research by combining two cross-cultural datasets—one of which was collected by our team—that have never before been integrated. This allows us to experimentally and systematically test the effects of three norm-based interventions on four dependent variables across 42 countries ($N = 16,089$) that vary in their degree of cultural tightness. While the main focus of this study is an overall assessment of whether the tightness of the cultural context moderates the effectiveness of social norm interventions, there are also differences among the three interventions that may interact differently with the cultural setting. Specifically, the interventions differ in the type of norm information they convey, and potentially the sociocultural mechanisms by which they exert influence. Broadly, normative information can convey the prevalence of behavior (i.e., descriptive norms), what is commonly approved of or valued (i.e., prescriptive injunctive norms), or what is disapproved of or regarded as taboo (i.e., proscriptive injunctive norms)[68]. Norm messages may be static—describing the status quo—or dynamic, reflecting changes in norms over time (i.e., dynamic norms)[31]. Additionally, some messages emphasize collective engagement (e.g., "working together" norms), while others merely describe what others do or think without suggesting coordinated action[69].

In the present research, we test three norm interventions that vary on the above dimensions. The Pluralistic Ignorance message asks participants to estimate the percentage of people in their country who believe that climate

change is a global emergency—implicitly invoking static descriptive and prescriptive injunctive norms. It then reveals the actual prevalence of people in the participant's country who believe climate change is a global emergency, thereby correcting systematic underestimation of existing social norms[70–72]. The Dynamic Norm message describes recent increases in public concern about climate change, support for climate policies, and approval of sustainable behaviors, thus conveying both dynamic prescriptive injunctive norms and descriptive norms[29,33,73]. Finally, the Work Together Norm message communicates strategies to reduce one's carbon footprint, while emphasizing the prevalence of such behaviors, combining both prescriptive and proscriptive injunctive norm information with descriptive norm information[69,74]. Together, these interventions reflect a range of norm manipulations used in the literature, allowing us to generalize across specific norm intervention design choices.

Despite potential differences in their underlying mechanisms, each intervention communicates evaluative information about social expectations—whether by highlighting what most people believe, emphasizing a shift in behavior, or signaling collective action. This range of norm information aligns with our primary goal: examining whether cultural tightness moderates the effectiveness of social norm interventions in general. Since cultural tightness is thought to broadly promote conformity to social norms and to inhibit deviance, we expect to see normative influences across behavioral and attitudinal outcomes. As such, we hypothesize that appeals to pro-climate social norms in tight cultural contexts would lead to greater climate belief (H1), support for mitigation policies (H2), intentions to share climate change information on social media (H3), and effortful mitigation behavior (H4) relative to looser cultural contexts.

These outcomes are important, in different ways, for broad-scale climate action. Belief in climate change is a well-established precursor to action—people are unlikely to change their behavior to mitigate climate change if they do not believe that climate change is real or human-caused[75,76]. Since policy support is critical for achieving structural change[77], and public support for climate policies strongly predicts whether a policy is passed into law[78], measuring policy support captures an important attitudinal precursor to systemic change. Willingness to publicly express support for climate action relates to social influence and the diffusion of pro-climate norms, which are central to processes of collective action[73], yet has the limitation of being a behavioral intention. The inclusion of the WEPT addresses this limitation, since it is a costly, incentivized behavioral measure[79], and expands the study to also include individual mitigation actions[80]. The norm interventions tested here provide information that can be used to update second-order beliefs (i.e., what people believe others think and do), and were thus designed to shift individual attitudes and behaviors via conformity to norms.

The interventions considered here have primarily been shown to be effective at shifting behaviors in the Global North, as such it is possible that they could have heterogeneous effects across cultural contexts given that they differ in terms of the norm referent group (e.g., neighborhood, nation, global region, the world), the type of social norm highlighted (e.g. injunctive, descriptive, dynamic), and which of the outcome variables they mention—we return to this possibility in the discussion. Nevertheless, while the three interventions vary in terms of the social norm information conveyed and how it is communicated, the primary goal of this work is not to attribute differences in moderation to the specific norm interventions in comparison to each other. It is, instead, to identify whether cultural context—and specifically its tightness or looseness—moderates the influence of norm information on attitudes and behaviors relative to a neutral control. Thus, our primary hypothesis is that the strength of different social norm interventions will be similarly moderated by cultural tightness. In exploratory analyses, we examine their effects individually and use these findings to suggest avenues for future work.

## Methods

To test the hypothesis that social-norm interventions have a greater impact in culturally tight relative to culturally loose contexts, we ran linear

regressions using data from a large-scale, cross-cultural experimental dataset ([81], ManyLabs dataset). This 63-nation study examined the impact of 11 experimental interventions (compared to a no-intervention control condition) on climate change beliefs, policy support, behavioral intentions, and mitigation behaviors. There were three different social norm interventions. Each participant was randomly assigned to one of the experimental conditions or the control condition, allowing us to examine whether the effect of each social norm manipulation differs systematically across cultural contexts. To understand the influence of cultural tightness on the efficacy of social norm interventions, we merged this data with a rank-order measure of each country's tightness[47]. We then conducted regressions to assess the interaction between cultural tightness and each of the three social norm interventions, compared to the control condition, for each of our primary dependent variables.

## Participants

The primary dataset contained 59,440 participants across 63 countries who were randomly assigned to one of 11 interventions or the control condition, completed all questions, and correctly answered both of the attention check questions (i.e., "*The color test you are about to take part in is very simple. Please select the color "purple" from the list below. We would like to make sure that you are reading these questions carefully.*" and "*In the previous section you viewed some information about climate change. To indicate you are reading this paragraph, please type the word sixty in the text box below.*"). For the purposes of our analyses, we excluded data from countries without a tightness-looseness estimate—that is, countries not captured in the dataset from Eriksson and colleagues[47]—which reduced our sample to 42 countries (see Table S2 for sample sizes per country). Additionally, we only included participants who were randomly assigned to the control condition or one of three social norm conditions, resulting in a total sample of 16,089 participants ($M_{age}$ = 38.8, $SD$ = 15.7; 50.3% female, 44.8% male, 0.7% non-binary or other). Table S3 in the Supplementary Materials reports the descriptive sample statistics by country. Broken down by condition, we had 4055 participants in the Dynamic Norm, 4030 in the Pluralistic Ignorance, 4030 in the Work-Together Norm, and 3974 in the control condition.

## Procedure

The data collection procedure was identical across countries and included obtaining local institutional ethics approval and translating study materials from English into local languages and back to English (for further details see ref. [82]). After providing informed consent, participants were randomly assigned to a between-subjects treatment group. Here, we analyze the data from the following four treatment groups: Dynamic Norm, Work-Together Norm, Pluralistic Ignorance, and a control. Participants in the control condition completed a task that was matched in terms of time and effort to the treatment conditions (i.e., they were asked to read an excerpt from *Great Expectations* by Charles Dickens that was unrelated to climate change) before completing all outcome measures. Participants in the three intervention conditions viewed text- and figure-based social norm messages (see Fig. 1) before completing all study outcomes. All participants were compensated for their time and were debriefed at the conclusion of the study.

## Social norm interventions

In the Dynamic Norm condition, adapted from Sparkman and Walton[29], participants were told that an increasing number of people in their country and globally are concerned about climate change and are taking action. They were presented with a figure showing change in concern in 11 countries, as well as the average, and were then told about actions that people are increasingly taking (see Fig. 1a). In the Work-Together Norm condition, adapted from Howe et al.[69], participants were shown a flyer stating that a majority of people are making efforts to reduce their carbon footprints and encouraging them to do the same through specific actions (see Fig. 1b). In the Pluralistic Ignorance condition, which was based on Geiger and Swim[70], participants were informed about a recent international public opinion

survey on climate change, known as the "Peoples' Climate Vote"[83], and were asked to estimate what percentage of people in their country—or region for countries in which national level information was not available—would agree that climate change is a global emergency. They were then told the real percentage of agreement in their country or region (see Fig. 1c). Research has found that individuals around the globe underestimate climate concern and policy support[84]. This intervention thus assumes that most participants underestimate the true extent of concern in their countries, and that the provision of the true estimates from The Peoples' Climate Vote's survey will, on average, reduce misperceptions about the norm. However, even for those who do not underestimate the norm, this intervention makes the norm salient.

## Dependent and independent variables

**Belief in climate change.** Participants responded to four items about their beliefs in the urgency of climate change (e.g., "*Climate change is a global emergency*" and "*Climate change poses a serious threat to humanity*") and reported how accurate they found each statement on a scale from 0 = "*not accurate at all*" to 100 = "*extremely accurate*". Participant responses were averaged into a composite score.

**Support for climate mitigation policies.** Participants were asked how much they supported nine policies (e.g., "*I support raising carbon taxes on gas/fossil fuels/coal*" and "*I support increasing the use of sustainable energy such as wind and solar energy*") on a scale from 0 = "*not at all*" to 100 = "*very much so*", with 50 = "*moderately*". Participant responses were averaged into a composite score.

**Willingness to share climate mitigation information on social media.** Participants were shown the message, "*Did you know that removing meat and dairy for only two out of three meals per day could decrease food-related carbon emissions by 60%?*". Participants were then asked whether they were willing to share the message on social media and could answer "*Yes, I am willing to share this information*" or "*I'm not willing to share that*". Participants were also given the option to report that they do not use social media, in which case they were excluded from analyses, resulting in a smaller sample size for this outcome.

**Engagement in a climate change mitigation task.** As a measure of online pro-environmental behavior, participants completed a multi-trial, web-based task that required cognitive effort in exchange for real monetary donations to an environmental organization (Work for Environmental Protection Task (WEPT)[85]). After completing a demo of the task, participants were given the opportunity to voluntarily engage in a number sorting task for up to eight trials. In each trial, participants were presented with a page of 60 two-digit numbers, and were asked to click on the numbers that consist of an even first digit (i.e., 2, 4, 6, 8) and an odd second digit (i.e., 1, 3, 5, 7, 9). Participants were informed that for each completed page of this task, the research team would donate one tree to the Eden Reforestation Project (https://www.edenprojects.org/). As per the guidelines in the original publication of this task, as well as the instructions given to participants, only pages where participants had a score of at least 80% contributed towards the number of planted trees, thus requiring them to exert real effort.

**Cultural tightness.** National estimates of cultural tightness are from a cross-cultural study[47] that contained country-level scores of cultural tightness for 42 of the 63 countries in our primary dataset. These scores were calculated based on the individual-level responses of 22,863 participants in 57 countries. Although primarily a student sample, Eriksson and colleagues[47] assessed the robustness of their measures by collecting data from additional non-student samples in 31 countries for comparison, and by collecting data from multiple student samples in different cities for 10 of the countries. Participants completed a six-item measure of cultural tightness[41], which assesses the degree to which social norms are

**(a) Dynamic Norm condition**

People in the United States and around the world are changing: **more and more people are concerned about climate change**, and are now taking action across *multiple fronts*.

**Since 2013, concerns about climate change have increased in most countries surveyed**

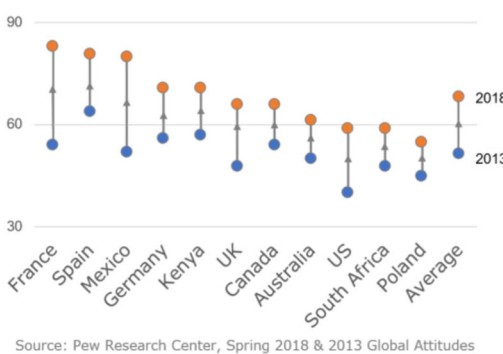

Source: Pew Research Center, Spring 2018 & 2013 Global Attitudes Surveys

What kinds of actions are people taking right now?

**More than ever before**, people are **making changes to their lifestyles**, **supporting policies to address climate change**, and are giving the issue **more time and attention**.

For example, more and more people from around the world are now:
• cutting back on personal consumption, especially meat and dairy products
• spending time, effort, and money on initiatives to mitigate climate change (for example, planting trees, offsetting carbon emissions)
• switching to low carbon modes of transportation (for example, taking bicycles)

There's also been a notable increase in support for climate change mitigation policy—some of the most popular policies include:
• attempting to conserve forests and land
• transitioning to solar, wind, and other renewable energy sources
• creating/raising carbon taxes on fossil fuels, coal, gas, etc.

**(b) Work-Together Norm condition**

Imagine you are seeing this flyer in your neighborhood. Please read it carefully!

You will be able to advance the page in 20 seconds.

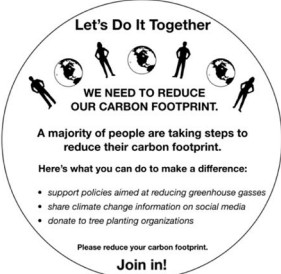

The flyer states:
"Let's do it together.
We need to reduce our carbon footprint.
A majority of people are taking steps to reduce their carbon footprint. Here's what you can do to make a difference:
- support policies aimed at reducing greenhouse gasses,
- share climate change information on social media,
- donate to tree planting organizations.
Please reduce your carbon footprint.
Join in!"

**(c) Pluralistic Ignorance condition**

Researchers recently conducted the "People's Climate Vote", which is the World's largest survey of public opinion on climate change ("global warming"). 1.2 million people completed the survey from 50 different countries around the globe. The survey included people from the United States.

**Think for a moment about Americans and their views on climate change. How many Americans do you think would agree with the statement "Climate change is a global emergency"?**

You estimated **[%] of Americans agree** that "Climate change is a global emergency."

In truth, the People's Climate Vote found **that 65% of Americans agree** that "Climate change is a global emergency."

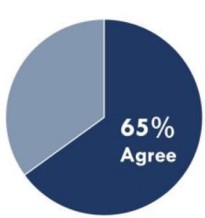

**Fig. 1 | Social norm intervention materials for United States participants.** Experimental materials for United States participants in the three intervention conditions: (**a**) Dynamic Norm, (**b**) Work-Together Norm, and (**c**) Pluralistic Ignorance. Dashed lines indicate page breaks separating different parts of the intervention.

pervasive, clearly defined, and reliably imposed within a given culture (e.g., *"People in this country almost always comply with social norms"* and *"In this country, if someone acts in an inappropriate way, others will strongly disapprove"*). Responses were measured on a six-point Likert scale, with scores ranging from 1 = *"strongly disagree"* to 6 = *"strongly agree"*, and were then averaged and standardized using a within-subject

standardization procedure to account for extreme responses and acquiescence bias across nations [41].

**Additional Covariates.** We included the following standard demographic control variables in our primary analyses: participants' age (*"How old are you?"*), gender (*"What is your gender?"*; response options

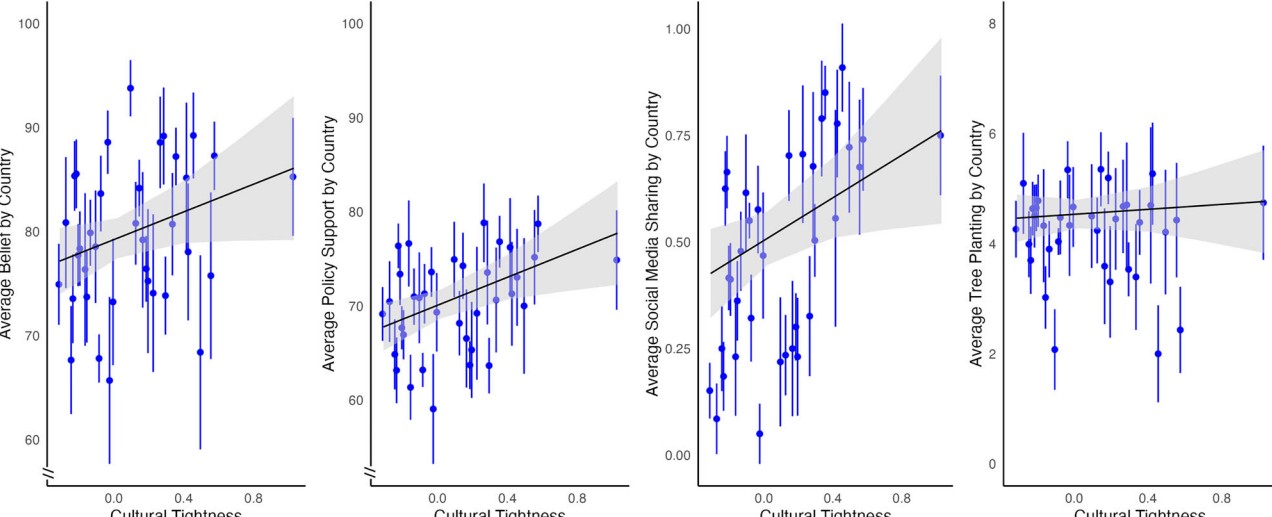

**Fig. 2 | Country-level associations between cultural tightness and dependent variables in the control condition.** Correlations between country-level means of each dependent variable and cultural tightness among participants in the control condition ($N = 3974$). Blue points represent country means, with lines indicating 95% confidence intervals within each country. Black lines and shaded regions show the linear trend and 95% confidence interval, estimated at the minimum and maximum quintiles of cultural tightness. All analyses include controls for age, gender, education, household income, political orientation, national emissions per capita, and GDP per capita.

were "*male*", "*female*", and "*non-binary/third gender/other*"), level of education ("*How many years of formal education have you completed?*"; coded as an ordinal scale with five responses options such as "*7-12 [up to high school]*"), household income ("*What is your total yearly family/household income?*"; coded with nine ordinal options such as "*$15,000 to $24,999*"), and political orientation for economic and social issues ("*What is your political orientation for social issues [e.g., health care, education, etc.]?*" and "*What is your political orientation for economic issues [e.g., taxes]?*" measured on a scale from 0 = "*Extremely liberal/left-wing*" to 100 = "*Extremely conservative/right-wing*" and averaged together). These individual-level covariates had significant correlations with all outcomes and cultural tightness, meriting their inclusion in the model (see Table S1). Furthermore, we also control for the following national-level variables: gross domestic product per capita (GDP[86]) and $CO_2$ emissions per capita accessed via Our World in Data [87].

### Analyses

All primary analyses were pre-registered on March 8th, 2024 (https://osf.io/ywcx6/), and all statistical analyses were performed using R Studio[88] and the R packages "lmer"[89], "lmerTest"[90], and "emmeans"[91]. To test our overall hypothesis that the efficacy of social norm interventions are moderated by cultural tightness, we ran a series of linear mixed-effects models for each of our dependent variables with experimental condition (entered as a series of dummy codes with the control as the reference group), country-level cultural tightness (grand-mean centered), and their interaction as our main independent variables of interest and random intercepts for each country. Our primary analyses included the standard individual and national controls described above. For all interaction effects, we also examined the difference between conditions at low (-1 *SD*) and high (+1 *SD*) levels of cultural tightness to describe the experimental effects under specific cultural conditions, i.e., tight (e.g., Singapore and India) and loose (e.g., Netherlands and Chile) contexts.

**Robustness analyses.** The analyses above provide a test of our primary hypotheses but have limitations that we address in robustness analyses. To ensure that the results were robust to different model specifications, we ran the following linear mixed-model regressions: (1) main analyses without controls; (2) main analyses controlling for country-level individualism-collectivism from the Culture Factor Group[92] to account for

the shared variance between these constructs and to test for unique moderating effects of cultural tightness (e.g.[93]) (see Fig. S1); (3) main analyses controlling for additional country-level measures (i.e., the Environmental Performance Index[94], the Vulnerability Index of Notre Dame Global Adaptation Initiative Country Index[95], the Gini Index[96], and the Human Development Index[97]) to further isolate cultural tightness from other country-level characteristics; and (4) main analyses excluding political orientation in order to include data from Kenya, which is missing this measure. Additionally, we (5) re-analyzed our main models using an alternative measure of cultural tightness measured by Uz ([98]; correlation with the measure by Eriksson et al.[47] is reported in Fig. S2) to ensure that observed effects were not specific to our measure. To examine whether there is sufficient evidence to interpret the null hypothesis as indicating a lack of moderation by cultural tightness, we (6) conducted a Bayesian analysis using models with a zero-one-inflated beta distribution for climate change belief, climate policy support, and tree-planting task, and with a Bernoulli distribution for intention to share climate mitigation information on social media. Finally, we also analyzed the Pluralistic Ignorance condition while accounting for (7) participants' baseline norm estimates and (8) whether their norm reference was the country or the larger geographic region (due to data availability). We did this to account for the possibility that tighter cultures are more accurate about the norm before the intervention since they have greater incentives to be well informed about the norm, as well as the possibility that norms about one's country are a more meaningful referent category than the region. All additional models are reported in the Supplementary Materials.

*A priori* sensitivity analyses using a random 10% of the data showed a minimum detectable effect size of $\eta_P^2 = .0005$ for 80% power to test H1 (as an example interaction). We thus determined that our models are well powered to test the hypothesized interactions.

### Results

#### Cultural context accounts for variation in climate attitudes and behavioral intentions

Cultural tightness was a significant predictor of three of our four primary outcome variables in the control condition (see Fig. 2). Controlling for individual and national level covariates, we found a significant positive association between cultural tightness and support for mitigation policies

**Table 1 | Results of linear mixed-effects model regressing climate change beliefs, policy preferences, and actions on social norm interventions**

| Predictors | Belief in Climate Change Estimates | Policy Support Estimates | Social Media Post Estimates | Tree Planting Estimates |
|---|---|---|---|---|
| Intercept | 77.96 *** (75.89–80.03) | 70.14 *** (68.32–71.95) | 0.56 *** (0.49–0.62) | 4.23 *** (3.99–4.46) |
| Dynamic Norm | 1.10 (−0.03–2.22) | 1.27 ** (0.34–2.19) | 0.07 *** (0.04–0.09) | −0.08 (−0.24–0.08) |
| Work-together Norm | −0.09 (−1.22–1.04) | 0.23 (−0.70–1.16) | 0.05 *** (0.03–0.08) | −0.37 *** (−0.53– −0.22) |
| Pluralistic Ignorance | 0.90 (−0.22–2.03) | 0.64 (−0.28–1.56) | 0.02 (−0.01–0.04) | -0.12 (−0.27–0.04) |
| Age | −0.09 *** (−0.12– −0.06) | -0.00 (−0.02–0.02) | −0.00 *** (−0.00– −0.00) | 0.04 *** (0.03–0.04) |
| Gender: Female | 4.02 *** (3.20–4.84) | 1.45 *** (0.78–2.12) | −0.05 *** (−0.07– −0.03) | 0.48 *** (0.36–0.59) |
| Gender: Non-binary/other | 4.33 (−0.84–9.51) | 3.61 (−0.66–7.87) | -0.10 (−0.22–0.01) | 0.42 (−0.31–1.15) |
| Education | 0.85 ** (0.20–1.49) | 1.40 *** (0.88–1.93) | 0.04 *** (0.02–0.05) | 0.06 (−0.03–0.15) |
| Income | 0.41 *** (0.17–0.65) | 0.38 *** (0.19–0.58) | 0.00 (−0.00–0.01) | 0.02 (−0.01–0.06) |
| Political orientation | −0.22 *** (−0.23– −0.20) | -0.14 *** (−0.15– −0.12) | 0.00 *** (0.00–0.00) | −0.01 *** (−0.01– −0.01) |
| Emissions per capita | −0.47 (−1.23–0.29) | −0.26 (−0.93–0.41) | −0.02 (−0.04–0.01) | −0.12 ** (−0.20– −0.04) |
| GDP per capita | −0.68 (−1.69–0.33) | −1.04 * (−1.93– −0.14) | −0.05 ** (−0.08– −0.01) | 0.02 (−0.09–0.13) |
| *Random Effects* | | | | |
| $\sigma^2$ | 520.29 | 349.63 | 0.20 | 10.40 |
| $\tau_{00}$ | 35.13 Country | 27.73 Country | 0.04 Country | 0.38 Country |
| ICC | 0.06 | 0.07 | 0.17 | 0.04 |
| N | 41 Country | 41 Country | 41 Country | 41 Country |
| Observations | 12760 | 12722 | 9794 | 12769 |
| Marginal $R^2$/Conditional $R^2$ | 0.065/0.125 | 0.045/0.115 | 0.054/0.210 | 0.046/0.080 |

*$p < 0.05$ **$p < 0.01$ ***$p < 0.001$

($b = 7.43$, 95% CI [2.13–12.74], $SE = 2.78$, $p = 0.01$), intentions to share climate information on social media ($b = 0.25$, 95% CI [0.04–0.46], $SE = 0.11$, $p = 0.03$), and belief in climate change ($b = 6.66$, 95% CI [−0.02–13.36], $SE = 3.51$, $p = 0.06$), though this effect was only marginally significant. Cultural tightness did not predict effortful tree planting ($b = 0.23$, 95% CI [−0.67–1.12], $SE = 0.47$, $p = 0.64$).

## Overall effect of norm interventions on climate attitudes and behaviors

Overall, we found mixed effects of the social norm interventions across both intervention type and outcome variable (see Fig. 1 for intervention materials). None of the three interventions significantly increased belief in climate change relative to the control condition, and the Dynamic Norm intervention was the only one to significantly increase support for mitigation policies compared to control (see Table 1). Participants in the Dynamic Norm and Work-Together Norm conditions were more willing to share climate-related information on social media compared to those in the control condition. The Work-Together Norm condition was the only intervention to impact behavior, but it *reduced* tree planting relative to the control condition. While all three interventions have demonstrated robust, reliable, and sometimes sizable effects on behavioral outcomes in U.S. samples[29,69,70], the results are less consistent when aggregating data across cultural contexts and in the domain of climate change, suggesting that the aggregate level effects may be masking important heterogeneity.

## Effect of norm interventions by cultural tightness

In the following sections, we describe the results of linear mixed-effects models with each dependent variable regressed on condition, cultural tightness, and the interaction of the two, resulting in four primary regression specifications. We report significant interaction terms from each of these models (see Table 2 for full results; see Fig. 3 for interaction effects). In exploratory analyses, we examine the differences among the three treatment conditions in tight and loose cultural contexts (±1 *SD*; see Fig. 4). For all interaction terms, we present uncorrected significance tests. Due to the risk of family wise error (4 dependent variables × 3 experimental condition comparisons), we also indicate whether results are robust to Bonferroni corrections.

**Belief in climate change.** Cultural tightness significantly and positively moderated the effect of the Work-Together Norm condition on climate change belief, compared to the control condition (see Table 2). Participants from tighter countries had higher belief in climate change in general (as seen in the control condition)—and the association between cultural tightness and belief was even stronger for those who received the Work-Together Norm message (see Fig. 4). However, contrary to our hypothesis, we found no interaction effect for Dynamic Norms and a significant *negative* interaction for Pluralistic Ignorance relative to the control condition (see Table 2). Looking at the effects in cultural contexts with high and low tightness, we found that respondents in the Pluralistic Ignorance condition did not differ from control at high levels of cultural

**Table 2 | Results of linear mixed-effects models regressing climate change beliefs, policy preferences, and actions on social norm interventions interacted with cultural tightness**

| Predictors | Belief in Climate Change Estimates | Policy Support Estimates | Social Media Post Estimates | Tree Planting Estimates |
|---|---|---|---|---|
| Intercept | 77.37 *** (75.32–79.42) | 69.56 *** (67.79–71.34) | 0.53 *** (0.47–−0.60) | 4.24 *** (3.99–4.48) |
| Cultural Tightness | 7.62 * (0.60–14.64) | 8.13 ** (2.05–14.22) | 0.27 * (0.06–0.49) | 0.04 (−0.79–0.88) |
| Dynamic Norm | 1.10 (−0.03–2.22) | 1.29 ** (0.36–2.21) | 0.07 *** (0.04–0.09) | -0.08 (−0.23–0.08) |
| Work-together Norm | −0.15 (−1.29–0.98) | 0.20 (−0.73–1.13) | 0.05 *** (0.03–0.08) | −0.37 *** (−0.53–−0.21) |
| Pluralistic Ignorance | 0.96 (−0.17–2.09) | 0.69 (−0.23–1.62) | 0.02 (−0.01–0.04) | −0.11 (−0.27–0.05) |
| Age | −0.09 *** (−0.12–−0.06) | −0.00 (−0.02–0.02) | −0.00 *** (−0.00–−0.00) | 0.04 *** (0.03–0.04) |
| Gender: Female | 4.03 *** (3.22–4.85) | 1.47 *** (0.80–2.13) | −0.05 *** (−0.07–−0.03) | 0.48 *** (0.36–0.59) |
| Gender: Non-binary/other | 4.39 (−0.78–9.56) | 3.67 (−0.60–7.93) | −0.10 (−0.22–0.01) | 0.42 (−0.31–1.15) |
| Education | 0.84 * (0.19–1.48) | 1.40 *** (0.87–1.93) | 0.04 *** (0.02–0.05) | 0.06 (−0.03–0.15) |
| Income | 0.42 *** (0.18–0.66) | 0.40 *** (0.20–0.59) | 0.00 (−0.00–0.01) | 0.02 (−0.01–0.06) |
| Political orientation | −0.22 *** (−0.23–−0.20) | −0.14 *** (−0.15–−0.12) | 0.00 *** (0.00–0.00) | −0.01 *** (−0.01–−0.01) |
| Emissions per capita | −0.22 (−0.97–0.53) | -0.02 (−0.68–0.63) | −0.01 (−0.03–0.02) | −0.12 ** (−0.21–−0.03) |
| GDP per capita | −0.54 (−1.51–0.42) | −0.90 * (−1.75–−0.06) | −0.04 * (−0.07–−0.01) | 0.02 (−0.09–0.13) |
| Dynamic Norm * Tightness | −0.21 (−4.74–4.32) | −1.74 (−5.46–1.98) | 0.01 (−0.09–0.11) | −0.03 (−0.67–0.61) |
| Work-together Norm * Tightness | 4.91 * (0.35–9.48) | 2.37 (−1.38–6.12) | 0.01 (−0.09–0.11) | −0.10 (−0.75–0.55) |
| Pluralistic Ignorance * Tightness | −5.39 * (−9.95–−0.83) | −4.67 * (-8.41 – -0.93) | −0.03 (−0.13–0.07) | −0.59 (−1.23–0.06) |
| *Random Effects* | | | | |
| σ² | 519.62 | 349.31 | 0.20 | 10.40 |
| τ₀₀ | 31.43 Country | 24.33 Country | 0.04 Country | 0.39 Country |
| ICC | 0.06 | 0.07 | 0.15 | 0.04 |
| N | 41 Country | 41 Country | 41 Country | 41 Country |
| Observations | 12760 | 12722 | 9794 | 12769 |
| Marginal $R^2$/Conditional $R^2$ | 0.067/0.121 | 0.04/0.109 | 0.066/0.203 | 0.047/0.082 |

*$p < 0.05$ **$p < 0.01$ ***$p < 0.001$

tightness ($b = -0.38$, 95% CI [$-1.94$–$1.17$], $SE = 0.79$, $p = 0.63$) but reported greater belief in climate change than the control group in loose cultures ($b = 2.30$, 95% CI [$0.67$–$3.94$], $SE = 0.84$, $p = 0.006$). These significant moderation effects were small and no longer significant after applying family-wise error corrections.

**Support for mitigation policies.** There was no significant interaction between cultural tightness and either the Dynamic Norm or Work-Together Norm conditions on policy support. Contrary to our hypothesis, we again found a *negative* interaction between cultural tightness and the Pluralistic Ignorance condition relative to the control group (see Table 2), though it was not robust to family-wise error correction. As with climate change belief, there were no significant differences in policy support between respondents in the Pluralistic Ignorance and control conditions at high levels of cultural tightness ($b = -0.47$, 95% CI [$-1.75$–$0.81$], $SE = 0.65$, $p = 0.47$), though respondents in the Pluralistic

Ignorance condition reported significantly higher levels of policy support relative to those in the control condition in culturally loose contexts ($b = 1.85$, 95% CI [$0.51$–$3.20$], $SE = 0.69$, $p = 0.006$).

**Posting on social media.** We found no significant interaction effects between cultural tightness and any of the social norm interventions on the intention to share climate mitigation information on social media ($p$s > .51; see Table 2).

**Tree planting.** There were no significant interactions between cultural tightness and either the Dynamic Norm or Work-Together Norm conditions on tree planting ($p$s > 0.76; see Table 2). Though only approaching significance ($p = 0.07$), we again found a *negative* association between the Pluralistic Ignorance condition and cultural tightness (see Table 2), mirroring the pattern observed for belief and policy support. At high levels of cultural tightness, the Pluralistic Ignorance condition resulted in

**Fig. 3 | Interactions between social norm interventions and cultural tightness across outcomes.**
Interaction effects between social norm intervention condition and cultural tightness on four outcomes: (**a**) belief in climate change (*N* = 12,760), (**b**) support for mitigation policies (*N* = 12,722), (**c**) willingness to share climate change information on social media (*N* = 9794), (**d**) and tree planting (*N* = 12,769). Lines represent the linear association between tightness and each outcome for each condition, with shaded areas indicating 95% confidence intervals. All analyses include controls for age, gender, education, household income, political orientation, national emissions per capita, and GDP per capita, and include random intercepts for country.

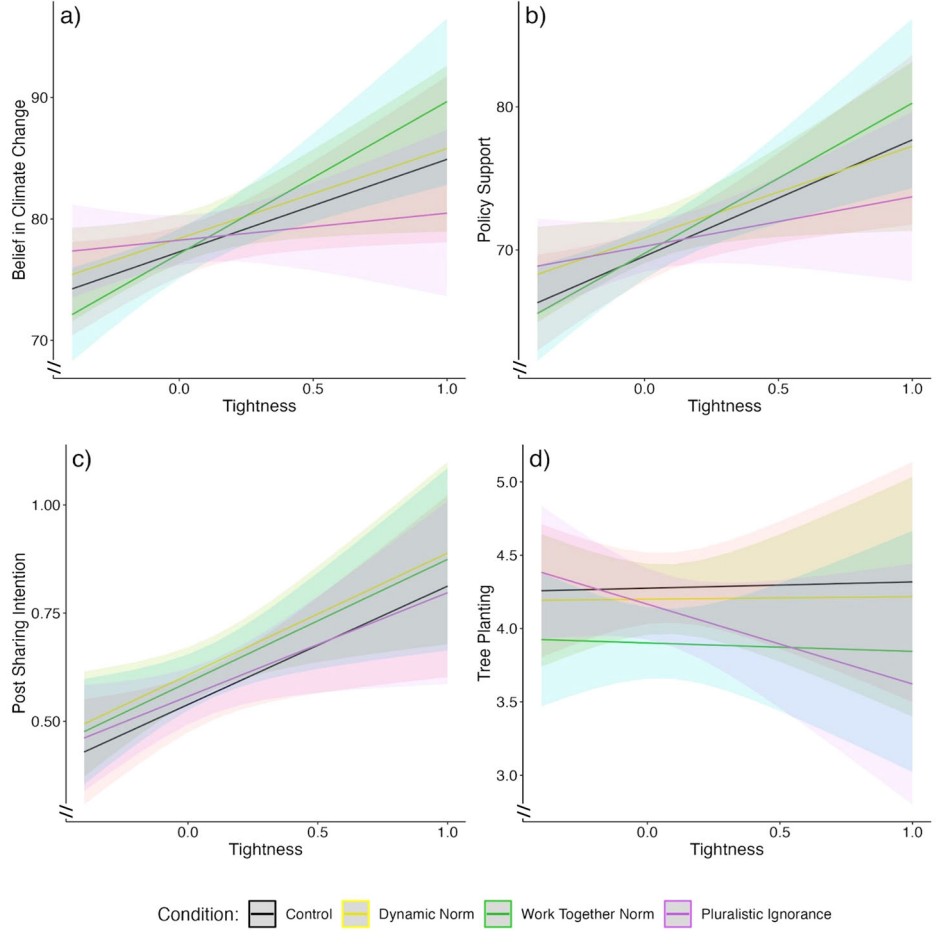

less tree planting compared to the control condition (*b* = −0.26, 95% CI [−0.48− −0.03], *SE* = 0.11, *p* = 0.02), but did not differ from control in culturally loose settings (*b* = 0.03, 95% CI [−0.19−0.27], *SE* = 0.01, *p* = 0.75).

### Robustness analyses

When excluding covariates or controlling for a country's individualism-collectivism score, we found modest differences (see Tables S4–S8). Cultural tightness no longer significantly moderated the relationship between the Work-Together Norm condition and belief in climate change, while it now significantly and negatively moderated the effect of the Pluralistic Ignorance condition on tree planting. Additionally, while mixed models are well-suited when random effects are collinear with the predictor variable, we also conducted a robustness check using a simple linear model with clustered standard errors by country. We found no major differences in results (see Tables S9–S12). Furthermore, when controlling for additional country-level variables (i.e., the Environmental Performance Index[94], the Vulnerability Index of Notre Dame Global Adaptation Initiative Country Index[95], the Gini Index[96], and the Human Development Index[97]; see Tables S22–S25), the results remained largely consistent with the main models reported in Table 2. The only difference concerns tree planting: in the primary analysis, the interaction between cultural tightness and the Pluralistic Ignorance intervention was significant only in the model without covariates, whereas in the supplementary analyses it was significant when we control for additional national-level covariates (see Table S25).

In the Pluralistic Ignorance condition, all participants were asked to give an estimate of their country's norm prior to being informed about that norm. Participants in the control condition were also asked to provide this estimate. While the exact percentage varied across countries, it is important

to note that the manipulation always described a majority view in that nation, with nearly all nations reporting a super-majority (i.e., more than 60% of the population). Nonetheless, because the actual norm varied by country, participants' estimates reflected differing distances from their national baseline. This is critical for normative influence: both the distance and direction of participants' misperceptions could introduce individual heterogeneity in the effect of the norm intervention—to the extent that the heterogeneity in accuracy varies across cultural context, this could impact our main inferences. To account for this, we calculated an "accuracy" score for each participant by calculating the difference between their estimate and the actual norm in their country/region. This difference score captures how close an individual is to the country-specific norm at baseline. This score was significantly associated with each outcome variable, indicating the importance of prior norm perceptions, but accounting for baseline estimates did not change the interaction effects (see Table S13), suggesting that our core results are robust to cross-national variation in norm estimates and the data provided by the intervention.

We also conducted two additional robustness analyses in the Pluralistic Ignorance condition. First, we ran these analyses using a binary indicator to distinguish respondents who over- or under-estimated the norm, since this may impact the efficacy of the Pluralistic Ignorance intervention. Specifically, the intervention was designed to correct systematic underestimation of existing social norms; therefore, for participants who overestimate the norm, learning that the norm is actually lower than they expected may actually have the opposite effect on the dependent variables. However, including this indicator did not change our results (see Table S14). Second, we ran these analyses controlling for the specific norm percentage displayed to participants in the Pluralistic Ignorance condition. For all outcomes, the inclusion of this covariate did not change our primary findings. In fact, while

**Fig. 4 | Main effects of social norm interventions at high and low cultural tightness.** Estimated main effects of the three social norm interventions on the four outcomes—belief in climate change ($N$ = 12,760), support for mitigation policies ($N$ = 12,722), willingness to share climate change information on social media ($N$ = 9794), and tree planting ($N$ = 12,769)—at high and low levels of cultural tightness (±1 $SD$). Horizontal error bars indicate 95% confidence intervals. Dotted and dashed vertical lines denote the control condition average at high and low cultural tightness, respectively.

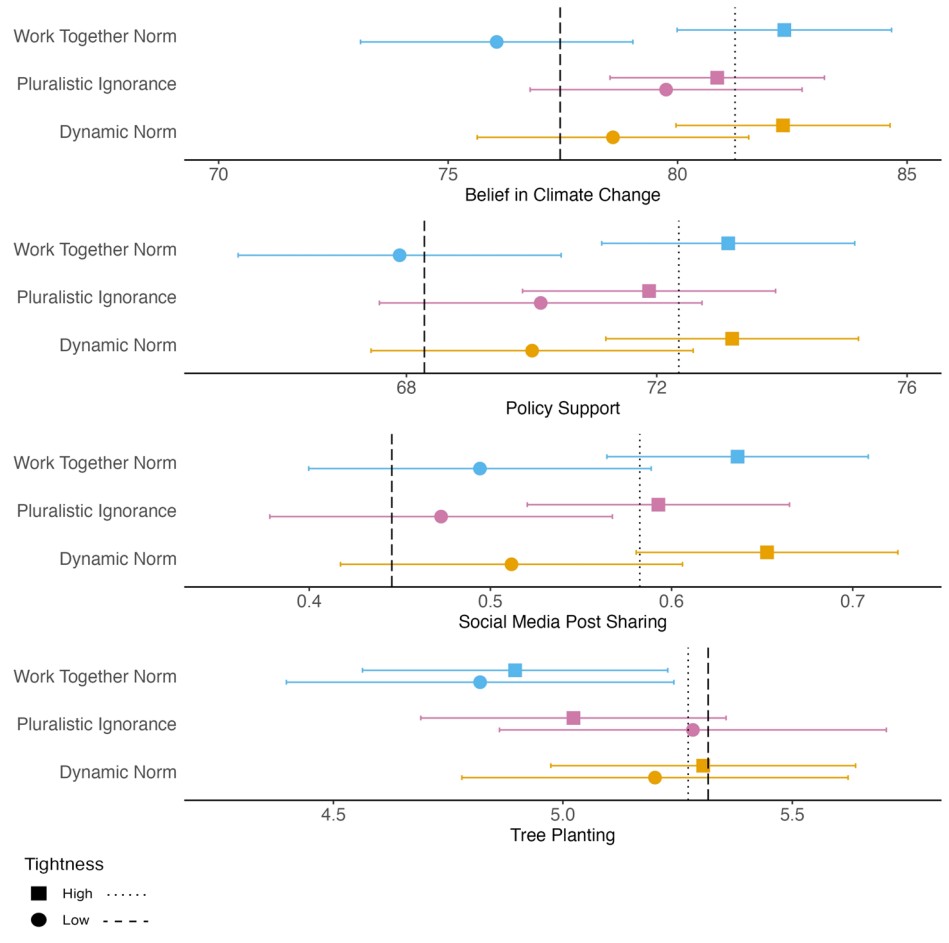

norm percentage was a significant predictor of willingness to post on social media, this effect was small and it was unrelated to all other outcomes (see Table S15).

Additionally, the intervention used data from the People's Climate Vote—a 50-country survey of public opinion on climate change—as an estimate of the percentage of people in the participant's country who believe climate change is a global emergency[83]; however, for countries without national-level results, participants were shown regional estimates. While participants were not told that these were regional rather than national-level estimates, if certain regions had greater variation in norms, participants in these regions may perceive the estimates to be less accurate and thus be less likely to conform to them. To account for this discrepancy, we included a dummy variable for countries where the norm referent was the region (vs. country) and found no differences in our results (see Table S16).

We replicated our primary analyses with an alternative measure of cultural tightness developed by Uz[98]. Unlike Gelfand and colleagues' measure[41], which aggregates individual self-reports of norm adherence and importance, Uz's index captures the amount of variance ($SD$) in values, norms, and behaviors within a culture using data from the European World Values Survey. Specifically, countries with greater variance in these measures are classified as looser, while those with lower variance are classified as tighter. This approach reduces subjectivity by measuring actual variation among participants rather than norm perceptions, and addresses concerns that aggregated self-reports may obscure meaningful within-country variability. Additionally, using this measure allows for the inclusion of different countries, as the datasets from Uz[98] and Eriksson et al[47]. differ in which and how many of the countries in our experimental data they cover (51 and 42 countries, respectively; see Models with an Alternative Cultural Tightness Measure in the Supplemental Materials for the list of countries gained and lost using this alternative measure).

For policy support, we found a significant negative interaction between this measure of cultural tightness and the Pluralistic Ignorance condition, but only in the model without covariates (see Table S18). We otherwise found no significant interactions for any of the other three outcome variables or social norm interventions (see Tables S17, S19–S20), providing additional evidence that the effectiveness of these interventions may not depend on cultural tightness.

Finally, we conducted our main analyses using Bayesian multilevel regression models to test whether our null effects provide evidence of the absence of an effect (see Bayesian Analysis Testing for Support of the Null Hypothesis in the Supplementary Materials for further details). Recall that in our mixed-effects linear models, we observed three significant interactions between cultural tightness and the social norm interventions: a positive interaction between cultural tightness and the Work-Together Norm condition on climate change belief, and significant *negative* interactions between cultural tightness and the Pluralistic Ignorance condition on belief and policy support. However, when conducting our main analyses using Bayesian models, we found consistent support for the null across almost all analyses. Specifically, we found *moderate* support for the null for the interaction between the Pluralistic Ignorance and tightness with policy support as the outcome ($BF_{01}$ = 4.44), which was significant in the mixed-effect model, and for all three interactions predicting message sharing ($BF_{01}$ range = 3.03 to 3.41). For all other interactions and outcomes we found *strong* support for the null, including when effects were significant in the mixed-effect linear models ($BF_{01}s$ > 11.24; see Table 3). These results suggest that the lack of significant interactions is not likely to be attributable to a lack of power, and instead indicates lack of variation in the effectiveness of social norm interventions by cultural tightness. However, these results should be interpreted with caution as some of these models showed only moderate support for the null.

**Table 3 | Bayes factors in support of the null hypothesis for the interactions between social norm conditions and cultural tightness**

| Intervention | Belief in Climate Change | Policy Support | Social Media Post | Tree Planting |
|---|---|---|---|---|
| Pluralistic Ignorance | 11.24 | 4.44 | 3.41 | 62.83 |
| Dynamic Norm | 84.58 | 64.91 | 3.03 | 22.45 |
| Work-Together Norm | 51.53 | 43.77 | 3.14 | 38.37 |

## Discussion

We provide a large-scale experimental assessment of whether cultural tightness moderates the impact of social norm interventions on climate attitudes and behaviors. Despite cross-sectional evidence that the importance of social norms in governing behavior across a host of domains varies predictably by cultural tightness, we find limited evidence that tightness moderates the influence of three common social norm interventions. We find heterogeneous effects across the different norm interventions and outcomes, suggesting that the relationship between conformity to social norm information and tightness is nuanced.

Specifically, we find that only one intervention—the Work-Together Norm—was positively moderated by cultural tightness, and for only one of four outcomes: climate change belief. Perhaps most notable, and contrary to our hypotheses, we find that the Pluralistic Ignorance condition has stronger effects on policy support and climate change belief in looser contexts. However, these patterns should be interpreted with caution as these significant interactions are the exception and are not robust to family-wise error corrections. Furthermore, *a priori* power analyses and a Bayesian analysis suggest that our null effects are not due to a lack of statistical power and provide robust evidence of the absence of an effect.

Although we do not find consistent evidence that tightness moderates the influence of social norm interventions, our results offer valuable insights into the effects of culture and interventions when considered independently. Our findings contribute to the existing body of correlational evidence demonstrating that climate attitudes and behaviors vary significantly across tight and loose cultures[59,60]. In particular, we find that tightness is positively correlated with support for climate mitigation policies and willingness to share pro-climate information on social media, which echoes past work showing a positive association between tightness and pro-environmental action[59]. In contrast, we do not find a significant correlation between cultural tightness and either belief in climate change or tree-planting. When aggregating across cultural contexts, we find little evidence that the social norm interventions reliably increase climate-related attitudes or behaviors. This lack of an effect is surprising given past evidence that these interventions are effective in increasing climate-related outcomes in the United States[29,69,70].

### Why did cultural tightness fail to moderate the efficacy of social norm interventions?

The interventions tested in the present work have been validated by prior studies, albeit primarily in the Global North (see, for example, Sparkman and Walton[29] for the Dynamic Norm intervention, Howe et al.[69] for the Work-Together Norm, and Geiger and Swim[70] for Pluralistic Ignorance). While the interventions were designed to be broadly applicable, they may not generalize across diverse cultural contexts. Indeed, all of the social norm interventions are effective in shifting at least some of our specified outcomes when limiting our sample to the United States (see Table S21), yet these effects do not generalize to the full cross-cultural sample. This suggests that the interventions may be particularly attuned to the United States and potentially other countries in the Global North context and less well-suited to countries in the Global South. Importantly, however, cultural tightness does not perfectly map onto distinctions between the Global North and South, so while this is a general concern for ManyLabs Megastudies, it does not represent a direct confound for the present study.

Some studies outside of the United States have found that dynamic norms, togetherness norms, and descriptive norm information can be effective means of shifting behaviors (e.g.[28,99,100]); however, these interventions may need to be culturally-attuned to each setting to be effective—including potentially delivered via different modalities beyond impersonal, light-touch, online surveys. This could suggest that ManyLabs Megastudies, while important for systematically testing interventions across cultural contexts, are not ideally suited to identify locally optimized and culturally-attuned interventions, especially since many of the tested interventions have been primarily validated in the Global North and proposed by scholars working in those settings. In this study, local samples were mostly collected by local researchers. However, the intervention text was submitted by experts who were largely based in the Global North. To better ensure that interventions are applicable and comparable across cultural contexts, future studies should pre-test and validate intervention materials. Additionally, interventions could be more strategically designed to theorize and test cross-cultural differences in the underlying mechanisms that make some interventions effective in one setting but not in others.

Furthermore, the interventions differ from each other in important ways, with potential implications for their hypothesized effectiveness across cultural contexts and for our interpretation of the results. For example, the Pluralistic Ignorance and Dynamic Norm interventions harness national-level norms, while the Work-Together intervention references norms among an unspecified referent group. One explanation for why we do not consistently find increased conformity among tighter cultures might be that broader or unspecified referent groups do not elicit a strong conformity response. Indeed, there is ample evidence that people are more likely to adhere to the social norms of relevant and often circumscribed referent groups—such as colleagues for work behaviors, or students for school behaviors—and this tendency may be especially important in tighter cultural contexts[101–103]. More circumscribed or different social milieus and interactions—such as specific locations, groups of people, online communities, or even interactions with AI—may vary in cultural tightness and moderate conformity to norm information. Indeed, large language models prompted with different languages (e.g. Chinese vs. English) appear to reflect culturally-resonant language (e.g. more interdependent social orientation when prompted in Chinese)[104]—future research could examine whether this content includes more references to social norms for languages associated with tighter cultures, and whether they are more effective when they do so.

Different social norm interventions are linked to distinct psychological mechanisms, which may vary in relevance or effectiveness depending on the cultural context. For instance, the Work-Together Norm intervention—which emphasizes an established group norm, highlights collective action, and invites individuals to 'join in'—has a stronger effect on climate beliefs in tighter cultures. This may be because it more clearly communicates social expectations through injunctive, rather than descriptive, language (e.g., "We need to reduce our carbon footprint") and offers direct behavioral guidance aligned with the outcome measures (e.g., "Donate to tree planting organizations"). Theories of cultural tightness pertain primarily to injunctive norms (i.e., explicit messages of what others should or should not do)[43–45]. Accordingly, the stronger effects of the Work-Together Norm in tight cultures may reflect its alignment with cultural sensitivity to prescriptive and proscriptive expectations.

In contrast, conformity to Dynamic Norm interventions requires individuals to align with emerging trends—attitudes or behaviors that are not yet established norms, but may become so in the future. In tighter cultures, where adherence to existing norms is stronger, people may be

slower to adopt emerging trends, or may even react negatively if they are perceived as conflicting with existing norms. Similarly, Pluralistic Ignorance interventions may be more effective in contexts where people are less aware of prevailing social norms. This may be more common in looser cultures, where behaviors and beliefs vary more, and where norm awareness matters less due to weaker social consequences for deviation[41]. In line with this, forthcoming work suggests that pluralistic ignorance about willingness to combat climate change is more pronounced in loose, as compared to tight, cultures[105].

It is worth noting that while injunctive norms can often be inferred from descriptive norm information[106], our interventions may not have conveyed clear social disapproval—a factor that individuals in tight cultures may be particularly sensitive to. As a result, interventions with more explicit proscriptive information might more closely align with existing theories of cultural tightness.

## Limitations

Several limitations should be considered when interpreting the results of the present research. First, although our sample is considerably larger and spans more countries than previous studies, it is constrained to the 42 countries included in Vlasceanu et al.'s[81] and Eriksson et al.'s[47] datasets. Some of these countries have relatively small sample sizes (see Table S2), and the Many-Labs dataset includes uneven sampling and heterogeneous representativeness across countries, reflecting differences in local data collection procedures and capacities. These factors should be considered when interpreting the results both within and across nations. While our dataset includes demographic quotas in many countries, not all countries' samples are fully representative of their population, and these deviations may limit the utility of national-level measures of cultural tightness. However, we do not expect that differences in representativeness are a confound here as we find no correlation between tightness and representativeness ($\rho = -0.21$, $p = 0.18$; see Representativeness Checks in the Supplementary Materials). Furthermore, representativeness is arguably less of a concern in our study because our design is experimental with randomization occurring within countries. Nonetheless, to account for differences in representativeness, we subset the sample to countries with low- vs. high-representativeness (see Tables S26–S29) and find only one significant interaction: cultural tightness and the Work-Together Norm interact to predict increased belief in climate change in the less-representative sample.

Second, our study examines three interventions selected based on suitability for an online study. This constraint precluded other promising interventions such as behavioral modeling of social norms (e.g., having celebrities or leaders behave in the targeted way[107–109]) or norm information about more socially-meaningful or geographically-proximal referent groups[25]. In addition, our interventions vary in the extent to which their content explicitly communicated injunctive and descriptive norms. A recent correlational study across eight Asian countries shows that injunctive norms are particularly influential in tight cultures[110]. While past research shows that descriptive norm information can shape beliefs about injunctive norms[29,106,111], our norm treatments vary along several dimensions, including whether they primarily emphasize injunctive or descriptive norms. Future research would therefore benefit from synchronized cross-cultural field experiments that systematically manipulate the components of the social norm interventions theorized to vary by cultural tightness.

Additionally, as Henrich and colleagues[112] argue, moving beyond WEIRD populations often requires not just a change in sampling frame but also substantial adaptation of research methodologies to ensure they are ecologically valid in the context of interest. Although effort was made to adapt study materials in consultation with local researchers, the interventions still reflect design choices rooted in U.S.-based studies, which may have reduced their effectiveness in some cultures. Furthermore, because this study was administered online, it is unlikely to have reached populations without internet access and familiarity with computer-based surveys; thus, our sampling frame systematically excludes certain populations—limiting

representativeness—while simultaneously bolstering survey construct validity. Future work could address this tradeoff by combining online methods with culturally resonant approaches and efforts to reach under-represented populations while preserving validity.

Third, although local researchers adapted study materials and recruited participants, it is possible that participants in some countries may have perceived the interventions as externally imposed. Even decentralized collaborations like this one may be interpreted through a colonial lens, particularly in historically exploited or economically disadvantaged settings. While this may influence the interpretation of and response to the materials, particularly in Global South countries, cultural tightness is distinct from whether a country is in the Global North or South and so we do not expect this influence to alter the results presented here. Nonetheless, future cross-cultural research should make efforts to ensure consistent sampling, the use of culturally resonant and validated materials, and participatory approaches to intervention and study design. These efforts can reduce the likelihood of potential confounds that impact the inferences that can be drawn about cultural differences. Additionally, while it is not possible to experimentally manipulate national levels of cultural tightness, future research might attempt to replicate the theorized interaction using pseudo-experimental designs to examine differences in conformity to norm interventions between people in the same cultural context who come from tight or loose backgrounds[66]. Our null results for national levels of cultural tightness do not imply that individual-level tightness would also yield null results, and testing this possibility directly remains an important next step for future research.

Fourth, three of our primary outcome variables rely on self-report data, making them potentially susceptible to demand effects and social desirability—both of which may vary across cultures. Our only behavioral outcome, the tree planting task (the WEPT), showed backfire effects across most interventions. This may reflect concerns that the tree-planting task may not generalize to real-world, high-impact environmental behaviors. Indeed, while some scholars argue that the WEPT is an effective measure of pro-environmental behavior[113,114], others have found that it only weakly relates to actual carbon footprints, suggesting that this task may not capture meaningful environmental impact[115]. Furthermore, although the WEPT is a costly measure of behavior that requires participants to take time and effort to plant more trees, thereby avoiding explicit monetary decisions, its use across countries with starkly different economic advantages may still raise concerns about comparability when time has opportunity costs and these may be different across country contexts and individuals. To the best of our knowledge, the WEPT had been validated only in a handful of countries (e.g., in Belgium,[85]; in the USA, and South Africa[114], and in the UK[114,115]), so it remains unclear to what extent economic background may influence performance across cultures. Additionally, it is possible that the task itself is unfamiliar and unintuitive for participants in many contexts. Future research could benefit from coordinated testing of social norm interventions in a field setting, using consequential behavioral outcomes or alternate incentivized behaviors, such as monetary measures calibrated to each country's GDP.

Finally, this study is focused on the domain of climate change. This domain may be unique for several reasons. Climate change is a global collective action problem that is highly polarized and marked by an uneven distribution of both impacts and responsibility for its causes[7]. Addressing climate change is often thought to require responses from economically powerful nations[116,117] and corporations, and individual action from wealthy, high-emitting individuals. Given these features, climate change may present a particularly difficult test case for the generalizability of norm interventions across cultures—especially insofar as cultural contexts also differ in their histories of industrialization and emissions. At the same time, climate change is a complex problem with many uncertainties that requires collective action to address, potentially rendering social norms an especially potent influence on behaviors and beliefs. Nevertheless, we did not find statistically significant evidence that historical emissions[118] and wealth[86] correlate with cultural tightness (see Table S39), nor do we observe clear and

consistent evidence that these national-level measures influence the main interaction results between tightness and the interventions (see Tables S30–S38). Future replications of this work in domains that are more relevant to a local or national context (e.g., public health) may therefore yield different results than those presented here.

## Conclusion

Overall, in a cross-cultural experimental examination of the moderating effect of cultural tightness on the effectiveness of social norm interventions, we find little evidence that tightness moderates the influence of three common social norm interventions. These results suggest that the relationship between cultural tightness and social norm information is nuanced, and likely sensitive to the design and content of the norm intervention. Our findings advance theory by suggesting that conformity to social norm messages is not uniformly amplified in tight relative to looser cultures. The present work points to the need for additional research to better understand the link between cultural tightness and social norm interventions, including how those interventions can be designed in culturally-attuned ways.

## Data availability

Both data sets are publicly available: https://osf.io/kp2zf/.

## Code availability

All code used to run analyses is publicly available: https://osf.io/kp2zf/.

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

## Acknowledgements
The authors received no specific funding for this work.

## Author contributions
Jane Acierno analyzed and visualized data and wrote the manuscript. Elisa Tedaldi and Joel Ginn analyzed data and wrote the manuscript. Danielle Goldwert facilitated access to the data and reviewed the manuscript. Madalina Vlasceanu acquired data and reviewed the manuscript. Sandra Geiger contributed to the conceptualization of the project and reviewed the

manuscript. Gregg Sparkman and Sara Constantino conceptualized the project, edited the manuscript, and provided supervision.

## Competing interests

The authors declare no competing interests.
