## [Transparent Peer Review file · Communications Psychology]

Conformity to social norm interventions is not amplified in tighter nations

Corresponding Author: Ms Jane Acierno

Version 0:

Decision Letter:

Dear Ms Acierno,

Thank you for your patience during the peer-review process. Your manuscript titled "Assessing the Role of Culture in Moderating Social Norm Interventions: A Global Experiment" has now been seen by 3 reviewers, and I include their comments at the end of this message. They find your work of interest but raised some important points. We are interested in the possibility of publishing your study in Communications Psychology, but would like to consider your responses to these concerns and assess a revised manuscript before we make a final decision on publication.

We therefore invite you to revise and resubmit your manuscript, along with a point-by-point response to the reviewers. Please highlight all changes in the manuscript text file.

Editorially, we consider it important that you provide more clarifications on the theoretical framework, methodological approach (particularly in cross-cultural hypothesis testing), and additional analyses as suggested by the reviewers. Greater integration of the studies findings with existing research as well as greater acknowledgement of the study's limitations is needed.

I am attaching an Editorial Requests Table that details critical reporting requirements for the revised manuscript. Please attend to each item and ensure your manuscript is fully compliant. If your revised manuscript is not aligned with these requests on major issues, such as those concerning statistics, it may be returned to you for further revisions without re-review.

Please submit the following items:

- Revised manuscript
- Point-by-point response to the referees' comments
- Cover letter (as a separate document)
- <https://www.nature.com/documents/nr-reporting-summary.pdf> Nature Research Reporting Summary
- Completed Editorial Request Table (attached).

via this link: Link Redacted .

Additional guidance is available in our style and formatting guide Communications Psychology formatting guide.

Best regards,

Yafeng Pan

Yafeng Pan, PhD
Editorial Board Member
Communications Psychology
orcid.org/0000-0002-5633-8313

REVIEWER EXPERTISE:

Reviewer #1: social norms,cultural tightness,large-scale survey

Reviewer #2: cultural tightness,large-scale survey

Reviewer #1: social norms,cultural tightness,large-scale survey

REVIEWER REPORTS:

Reviewer #1 (Remarks to the Author):

This study represents a commendable and ambitious effort to experimentally test the moderating role of cultural tightness in the effectiveness of social norm interventions across 42 countries—a significant contribution to the literature on cross-cultural behavioral science. The research question is theoretically grounded, addressing a critical gap in understanding how norm-based interventions might need to be tailored to different cultural contexts. The large-scale, multi-national design is a major strength, as is the inclusion of diverse outcome measures (beliefs, policy support, behavioral intentions, and actual behavior). The authors also employ rigorous analytical methods, including Bayesian approaches to interpret null effects.

However, the study has several limitations that temper the robustness and generalizability of its conclusions. The heterogeneous and often non-representative samples across countries raise concerns about whether the findings truly reflect population-level cultural differences. The interventions, while well-validated in Western contexts, may not fully account for cultural nuances in norm perception and adherence. Additionally, the mixed and often null results suggest that the relationship between cultural tightness and norm intervention efficacy may be more complex than hypothesized, requiring deeper theoretical and methodological exploration.

Below, I provide specific, actionable critiques to address these issues and strengthen the manuscript's impact.

- 1. Sample Representativeness and Generalizability.** The study includes 42 countries, but the samples are uneven in size and demographic composition (e.g., some countries have small or non-representative samples). The authors should explicitly discuss how this affects the generalizability of findings, particularly for tight cultures where norm adherence may vary by subgroup (e.g., urban vs. rural populations). A sensitivity analysis comparing effects across high- vs. low-representativeness samples would strengthen claims.
- 2. Intervention Content Validity Across Cultures** The norm interventions were developed and tested primarily in the U.S. (Global North). The manuscript should address whether the wording, framing, or norm referents (e.g., "people in your country") might have different connotations in tight vs. loose cultures. For example, collective-action appeals (Work-Together Norm) could inadvertently trigger reactance in individualistic-but-tight cultures. A qualitative pre-test of intervention materials in select cultures would help justify cross-cultural applicability.
- 3. Null Results and Bayesian Analysis Interpretation** While Bayesian analyses support the null for most interactions, the manuscript does not sufficiently explore alternative explanations for the lack of moderation. For instance, could

measurement error in cultural tightness (e.g., using country-level aggregates) obscure individual-level effects? The authors should explicitly test whether within-country variation in tightness (if available) predicts intervention efficacy better than national scores.

4. Behavioral Measure Limitations The tree-planting task (WEPT) showed backfiring effects and weak ecological validity (as noted in L90–91). The authors should clarify whether this reflects a true intervention effect or a task artifact (e.g., perceived low impact of tree-planting in certain cultures). Adding a validity check (e.g., participant perceptions of the task's meaningfulness) or comparing results to a non-climate behavioral measure would help disentangle this.

5. Covariate Control and Confounding The models control for GDP and CO₂ emissions, but other national-level confounders (e.g., climate vulnerability, political stability) may co-vary with tightness and influence outcomes. The authors should justify their choice of covariates and consider adding robustness checks with additional controls (e.g., Environmental Performance Index scores) to rule out omitted variable bias.

Other points:

P9, L198–206: The interaction effects for Pluralistic Ignorance are highlighted but not robust to corrections. The discussion should explicitly caution against overinterpreting these patterns without replication.

P17, L335–342: The speculation about climate change being a "unique domain" needs empirical support. Cite studies showing how norm interventions perform differently in climate vs. other collective action problems (e.g., public health).

Methods (P23, L473): Clarify how "representative samples" were defined (e.g., quotas, random sampling) and whether this aligns with standard practices in cross-cultural research.

These critiques target gaps where additional analyses or discussion could substantially strengthen the paper's contributions.

Reviewer #2 (Remarks to the Author):

This is a review of the manuscript entitled "Assessing the Role of Culture in Moderating Social Norm Interventions: A Global Experiment," submitted to *Nature: Communications Psychology*. To begin, the motivation to understand cultural variance in the effectiveness of different norm-based interventions is laudable, and beginning with cultural tightness is reasonable. There are also several strengths to the paper, including the large sample size and large number of sample locations, as well as the strong experimental control toward the goal of examining replicability. However, I also have several concerns about the methods for the cross-cultural hypothesis testing in the manuscript and the conclusions drawn, which are the focus below.

1. Moving beyond cultures of the Global North is a motivation for this paper, and is in some ways handled effectively (e.g., many cultures analyses; appreciating that the Global North disproportionately causes the global climate crisis, while the Global South disproportionately suffers the consequences of it). However, a deep consideration of what this means for the research is lacking. Henrich, Heine, and Norenzayan (2010) present analysis that informs how the goals of this project as articulated may be too mismatched from the methods. Specifically, Henrich et al. (2010) examine how the research methodologies themselves can be too far removed from cultural contexts of interest, and they underscore how movement away from US samples to contexts that are less Western, Educated, Industrialized, Rich, and Democratic (WEIRD) often requires significant adaptation of research methodologies. The manuscript Discussion section almost acknowledges this in describing normative influence in Asia and how tight cultures require more proximal norm information, but consideration of the global generalizability of the methods and assumptions about methodological fit for the samples require attention. I unpack this more below in points 2, 6, and 8.
2. It is not clear in the manuscript, supplemental materials, or original dataset publication precisely who the participants were and how translations were managed. While this matters less for the original publication that was not focused on potential cultural differences, to draw conclusions about cultural variance – particularly about how tightness leads to reduced normative influence – requires this was all done to the highest standards.
3. The experimental manipulations confound the type of normative influence with other factors, including the presence of quantitative data and explicit requests to take actions later measured in dependent variables. This makes it challenging to disentangle the effects of the intended normative influence relative to other confounded factors.
4. The numbers represented in the quantitative data for normative and pluralistic ignorance manipulations differed for each country. This is important for normative influence, but as the norm setting data shift as we move away from the US, the influence patterns should too – moving toward the norm will be different when the norm is different.
5. The manuscript also acknowledges how the dynamic norm and pluralistic ignorance experimental conditions focused on the most proximal data available (e.g., Kenya would be included for Kenyan participants), but it is unclear which comparison countries were included in each country's stimulus materials. Given the disproportionate responsibility and consequences for different countries, the comparison groups could strongly influence participants in many ways.
6. Further, it is unclear if the cultural variance or heterogeneity between (and within) samples is due to differences in tightness or to methodological artifacts or to histories of colonization, which also relates to the quality of translations and the presentation of the research to participants. If the study materials read as if created by foreigners (which would be very reasonable given that they were for most countries), this can be experienced with a shadow of colonial history by

participants, especially in the most economically and environmentally disadvantaged countries. Many of these countries have been most exploited by formal colonization and business and governmental extraction by the Global North. The resistance to influence and even reactance could be based on local efforts to reclaim control, which would require emic considerations of how these interventions might be experienced in the cultural contexts coded as tight in this research. Even assuming that materials were translated brilliantly, information about foreign researcher leads who are Western or more economically powerful within continents could still trigger such resistance.

7. The focus on environmental sustainability introduces more concerns about the methodological fit for participants. For example, while gross domestic product per capita and income were covariates, neither were tested as interacting with the interventions. These analyses are both important for the contribution relative to disproportionate environmental crisis responsibility and consequences as articulated in the manuscript.

8. As a smaller point, the computer numeracy work for planting trees likely feels much stranger to participants who are extremely economically disadvantaged and in less industrialized and economically developed contexts.

In closing, again, scale of data collection and importance of the research questions deserve recognition. I hope my suggestions help the authors to develop the paper to its potential, whether eventually published at Nature Communications Psychology or elsewhere.

Reviewer #3 (Remarks to the Author):

This work examines the role of cultural tightness in moderating the effectiveness of social norm interventions aimed at promoting climate change attitudes and behaviors. The topic is very interesting. Detailed comments are provided below.

1. Theoretical Framework. The authors present a reasonably comprehensive overview of the literature on cultural tightness and looseness. However, the rationale for including the specific set of social norm persuasion strategies – the pluralistic ignorance norm message, the dynamic norm message, and the Work Together Norm message – requires further clarification. What are the key distinctions between these three types of persuasive information, and what are the a priori hypothesized relationships between them in the context of cultural tightness? The study would benefit from a clearer articulation of the unique mechanisms by which each norm message is expected to influence individuals in cultures with varying degrees of tightness.

Furthermore, the justification for focusing on climate belief (H1), support for mitigation policies (H2), intentions to share climate change information on social media (H3), and effortful mitigation behavior (H4) is not sufficiently substantiated. Why were these specific outcome variables selected, and what theoretical rationale connects them to the social norm interventions and cultural tightness? A more robust theoretical foundation, supported by existing literature, is needed to explain the choice of these outcomes. Are these outcomes intended to capture attitude change, behavioral intention, actual behavior, or all three? Clarifying the relationships among these different levels of outcomes would strengthen the theoretical framework.

2. Methodology. The methodology section presents a design involving three different social norm interventions, each underpinned by distinct psychological mechanisms. The materials used in these interventions appear to differ substantially in both content and structure. A more detailed description and justification of these differences is needed. However, the control condition seems rudimentary, consisting of "reading a non-related text from Great Expectations by Charles Dickens." What specific construct(s) was this control condition designed to control for?

3. Covariates. While the authors controlled for some individual and group-level variables, further consideration should be given to a broader range of potentially confounding factors. Specifically, the following country-level variables might be important to include as covariates: urbanization, Gini, proportion of Environmental Organizations or pollution levels. These factors may be correlated with both the interventions and individual attitudes towards climate change, potentially confounding the observed relationships.

4. Results. The authors report that "Cultural tightness predicts cross-cultural differences in three of our four primary outcome variables in the control condition." This finding warrants further investigation. Why was this correlation calculated for the control condition? Why wasn't the correlation also examined for the experimental conditions? The authors should report the correlations in all conditions and justify the reason behind this.

5. Intervention Effects and Inconsistencies. A critical issue of the study is that none of the three interventions significantly increased belief in climate change relative to the control condition, and the Dynamic Norm intervention was the only one to significantly increase support for mitigation policies compared to control. How does this finding align with previous research on social norm interventions and climate change? Do other studies find similar null effects, or are the results contradictory? A discussion of the consistency (or inconsistency) with the existing literature is essential.

Furthermore, the interaction effects between cultural tightness and the interventions are inconsistent across different outcome variables. These inconsistent results highlight the complexity of social norm persuasion and the potentially unstable nature of cultural values as moderators. While the authors conducted robustness analyses, these inconsistencies persist. The most pressing challenge may be to address the differences between the interventions and outcomes, and to investigate the underlying mechanisms that might explain these variations. It could also be helpful to determine if the different interventions target different levels of the same variable.

6. Alternative model. An intriguing alternative to consider is whether cultural tightness might function more appropriately as an IV, with the different intervention strategies acting as moderators. This approach could provide a different perspective on how cultures independently influence behavior, and how the impacts of these behaviors can be moderated by outside influences.

7. Theoretical Contribution. Overall, the researchers should carefully consider the theoretical contribution of their findings to the broader literature on social norm interventions and cultural psychology. What new insights does this study offer for understanding the complex interplay between culture, social influence, and climate change attitudes and behaviors?

Version 1:

Decision Letter:

Dear Ms Acierno,

Your manuscript titled "Assessing the Role of Culture in Moderating Social Norm Interventions: A Global Experiment" has now been seen by our reviewers, whose comments appear below. In light of their advice I am delighted to say that we are happy, in principle, to publish a suitably revised version in Communications Psychology.

We therefore invite you to revise your paper one last time to address the remaining concerns of our reviewers and a list of editorial requests. At the same time we ask that you edit your manuscript to comply with our format requirements and to maximise the accessibility and therefore the impact of your work.

EDITORIAL REQUESTS:

SUBMISSION INFORMATION:

OPEN ACCESS:

Communications Psychology is a fully open access journal. Articles are made freely accessible on publication. For further

information about article processing charges, open access funding, and advice and support from Nature Research, please visit <https://www.nature.com/commpsychol/open-access>

* **DATA AVAILABILITY:**

Link Redacted

Best regards,

Jennifer Bellingtier

Jennifer Bellingtier, PhD
Senior Editor
Communications Psychology

Yafeng Pan, PhD
Editorial Board Member
Communications Psychology
orcid.org/0000-0002-5633-8313

REVIEWERS' EXPERTISE:

Reviewer #1: social norms,cultural tightness,large-scale survey
Reviewer #2: cultural tightness,large-scale survey

REVIEWERS' COMMENTS:

Reviewer #1 (Remarks to the Author):

The author's revisions have significantly improved the clarity of the article. From the perspective of further enhancing the significance of this paper for the future development of human society, I have one additional point I would like to discuss with the author. Current research tends to focus on validating the effects of social norm interventions across different cultures within a broader range of countries. However, when discussing cultural differences, in addition to spatial variations (such as regional differences), temporal differences—that is, cultural evolution—are also crucial. Contemporary society is on the brink of a profound cultural transformation: the transition into a cultural model of deep human-AI integration. Compared to cultural groups composed solely of humans, this new form of society, characterized by human-AI integration, will witness significant changes in social and cultural norms. Recent studies have begun to reveal the cultural values of AI, as well as the cultural stereotypes and biases it may introduce. In this context, how might this new cultural landscape relate to the effects of these

three social norm interventions, and what are the author's perspectives on this? A discussion on this topic could further enhance the interdisciplinary impact of this paper.

Ref:

(2025). Cultural tendencies in generative AI. *Nature Human Behaviour*, 1-10.

(2025). The cultural stereotype and cultural bias of ChatGPT. *Journal of Pacific Rim Psychology*, 19, 18344909251355673.

(2024). Cultural bias and cultural alignment of large language models. *PNAS nexus*, 3(9), pgae346.

Reviewer #2 (Remarks to the Author):

Thank you to the author team for your excellent improvements to the manuscript!

The clarifications in the response letter and in the manuscript itself dramatically improve my confidence in the conclusions drawn in this paper. Further, the additional analyses were both thorough and accurately interpreted. This is an impressive transformation.

Reviewer #1

This study represents a commendable and ambitious effort to experimentally test the moderating role of cultural tightness in the effectiveness of social norm interventions across 42 countries—a significant contribution to the literature on cross-cultural behavioral science. The research question is theoretically grounded, addressing a critical gap in understanding how norm-based interventions might need to be tailored to different cultural contexts. The large-scale, multi-national design is a major strength, as is the inclusion of diverse outcome measures (beliefs, policy support, behavioral intentions, and actual behavior). The authors also employ rigorous analytical methods, including Bayesian approaches to interpret null effects.

Thank you for your comments on the impact and importance of this work.

However, the study has several limitations that temper the robustness and generalizability of its conclusions. The heterogeneous and often non-representative samples across countries raise concerns about whether the findings truly reflect population-level cultural differences. The interventions, while well-validated in Western contexts, may not fully account for cultural nuances in norm perception and adherence. Additionally, the mixed and often null results suggest that the relationship between cultural tightness and norm intervention efficacy may be more complex than hypothesized, requiring deeper theoretical and methodological exploration.

Below, I provide specific, actionable critiques to address these issues and strengthen the manuscript's impact.

1. Sample Representativeness and Generalizability. The study includes 42 countries, but the samples are uneven in size and demographic composition (e.g., some countries have small or non-representative samples). The authors should explicitly discuss how this affects the generalizability of findings, particularly for tight cultures where norm adherence may vary by subgroup (e.g., urban vs. rural populations). A sensitivity analysis comparing effects across high- vs. low-representativeness samples would strengthen claims.

We appreciate the reviewer's thoughtful comment regarding variation in sample sizes and demographic composition, and how these factors may affect the generalizability of our findings. We conducted additional analyses to better address this potential concern. First, we tested whether cultural tightness correlated with representativeness, measured in terms of how many demographic factors in the sample matched national benchmarks (see section "Representativeness Checks" in the Supplemental Materials). Fortunately, these variables are not correlated ($\rho = -.21$, $p = .18$), suggesting that representativeness varies independently of tightness and thus is unlikely to introduce a confound.

However, as an added precaution, we conducted a sensitivity analysis in which we compared the effects between countries with more-representative samples (i.e., countries with

samples that were representative on at least two demographic characteristics) and countries with less-representative samples. When conducting these analyses on the subgroups of more- and less-representative samples, we found that tightness significantly interacted with the Work-Together Norm intervention to predict Belief in Climate Change in the less-representative sample, but not in the more-representative sample. However, this was the only difference in significance between the two samples, and our other main effects remained consistent. As such, it does not appear that sample representativeness changes the hypothesized moderation effects across the norm interventions. We now report these results in Tables S27-S30 of the Supplementary Materials, and address this point explicitly in our new limitations section:

“We do not expect that differences in representativeness are a confound here as we find no correlation between tightness and representativeness ($\rho = -.21$, $p = .18$; see Representativeness Checks in the Supplemental Materials). Furthermore, representativeness is arguably less of a concern in our study because our design is experimental with randomization occurring within countries. However, to account for differences in representativeness, we subset the sample to countries with low- vs. high-representativeness (see Tables S27-S30) and find only one significant interaction: cultural tightness and the Work-Together Norm interact to predict increased belief in climate change in the less-representative sample.”

2. Intervention Content Validity Across Cultures The norm interventions were developed and tested primarily in the U.S. (Global North). The manuscript should address whether the wording, framing, or norm referents (e.g., "people in your country") might have different connotations in tight vs. loose cultures. For example, collective-action appeals (Work-Together Norm) could inadvertently trigger reactance in individualistic-but-tight cultures. A qualitative pre-test of intervention materials in select cultures would help justify cross-cultural applicability.

While we can no longer pre-test the intervention materials since the study has already been conducted, we have expanded the Introduction to include this important point. We now note that the wording, norm framing, and referent may function differently across cultures.

“The interventions considered here have primarily been shown to be effective at shifting behaviors in the Global North, as such it is possible that they could have heterogeneous effects across cultural contexts given that they differ in terms of the norm referent group (e.g., neighborhood, nation, global region, the world), the type of social norm highlighted (e.g. injunctive, descriptive, dynamic), and which of the outcome variables they mention—we return to this possibility in the discussion.”

Additionally, in the Discussion we now mention context-specific pre-testing of intervention materials as an important direction for future research:

“In this study, local samples were mostly collected by local researchers. However, the intervention text was submitted by experts who were largely based in the Global North. To better ensure that interventions are applicable and comparable across cultural contexts, future studies should pre-test and validate intervention materials.”

3. Null Results and Bayesian Analysis Interpretation While Bayesian analyses support the null for most interactions, the manuscript does not sufficiently explore alternative explanations for the lack of moderation. For instance, could measurement error in cultural tightness (e.g., using country-level aggregates) obscure individual-level effects? The authors should explicitly test whether within-country variation in tightness (if available) predicts intervention efficacy better than national scores.

We thank the reviewer for this thoughtful suggestion. First, we would like to be clear that null results for national levels of cultural tightness do not mean that individual levels of tightness would also necessarily have null results. We have made this point more prominent in the manuscript, especially in the discussion section in which we state: *“Our null results for national levels of cultural tightness do not imply that individual-level tightness would also yield null results, and testing this possibility directly remains an important next step for future research.”*

Without individual measures of cultural tightness from each participant within each country, we can only address this issue by using an alternative measure of tightness and clarifying the distinction between these measures. We have thus also revised the paragraph discussing our use of an alternative measure of cultural tightness involving within-country variation—specifically, one that measures variance in norms, values, and behaviors—to more directly address concerns about measurement and the distinction between conceptualizations of tightness. We now explicitly note that Uz’s measure was used to address the possibility that aggregation of individual self-reports might obscure meaningful variability. Further, we reiterate that although this approach captures a different facet of cultural tightness and includes a different set of countries, the overall pattern of results remains relatively unchanged: we do not find evidence for moderation effects. This consistency suggests that the null findings from our primary measure are not an artifact of the way that cultural tightness was operationalized.

“We replicated our primary analyses with an alternative measure of cultural tightness developed by Uz [100]. Unlike Gelfand and colleagues’ measure [41], which aggregates individual self-reports of norm adherence and importance, Uz’s index captures the amount of variance (SD) in values, norms, and behaviors within a culture using data from the European World Values Survey. Specifically, countries with greater variance in these measures are classified as looser, while those with lower variance are classified as tighter. This approach reduces subjectivity by measuring actual variation among participants rather than norm perceptions, and addresses concerns that aggregated self-reports may obscure meaningful within-country variability. [...] For policy support, we find a significant negative interaction between this measure of cultural tightness and the Pluralistic Ignorance condition, but only in the model without covariates (see Table S18). We otherwise find no significant interactions for any of the other three outcome variables or social norm interventions (see Tables S17, S19-S20), providing additional evidence that the effectiveness of these interventions may not depend on cultural tightness.”

4. Behavioral Measure Limitations The tree-planting task (WEPT) showed backfiring effects and weak ecological validity (as noted in L90–91). The authors should clarify whether this reflects a true intervention effect or a task artifact (e.g., perceived low impact of tree-planting in certain cultures). Adding a validity check (e.g., participant

perceptions of the task’s meaningfulness) or comparing results to a non-climate behavioral measure would help disentangle this.

We agree that there are limitations with the tree-planting task and these might explain the backfire effects. Unfortunately, the study did not include a validity check or a non-climate behavioral measure so we are unable to directly test this limitation. This measure was originally included because it reflects an easily implemented behavioral measure of environmentalism that could be used across countries and circumvents concerns associated with other commonly used behavioral measures (e.g., donation behavior) that involve financial commitments. The validity of this task has been established in other studies, though it is not without its critics, who argue that this task does not capture meaningful environmental behavior. We now cite this criticism in our limitations section.

“[...], others have found that it only weakly relates to actual carbon footprints, suggesting that this task may not capture meaningful environmental impact [114]”

Additionally, we note other limitations with this measure, such as the fact that it has not been validated across our country contexts. As it is a newer measure, it is possible that interpretation and performance of the task may not equally reflect pro-environmental behavior in all cultures, particularly as it may relate to wealth.

“Furthermore, although the WEPT is a costly measure of behavior that requires participants to take time and effort to plant more trees, thereby avoiding explicit monetary decisions, its use across countries with starkly different economic advantages may still raise concerns about comparability when time has opportunity costs and these may be different across country contexts and individuals. To the best of our knowledge, the WEPT had been validated only in a handful of countries (e.g., in Belgium, [87]; in the USA, and South Africa [113], and in the UK, [113-114]), so it remains unclear to what extent economic background may influence performance across cultures. Additionally, it is possible that the task itself is unfamiliar and unintuitive for participants in many contexts.”

5. Covariate Control and Confounding The models control for GDP and CO₂ emissions, but other national-level confounders (e.g., climate vulnerability, political stability) may co-vary with tightness and influence outcomes. The authors should justify their choice of covariates and consider adding robustness checks with additional controls (e.g., Environmental Performance Index scores) to rule out omitted variable bias.

We originally only controlled for GDP and CO₂ emissions to avoid possible overfitting or collinearity issues. However, we acknowledge that there may be other factors that relate to our outcomes of interest. Based on your comments, as well as feedback from our other reviewers, we have added robustness checks to our Supplemental Materials including models with the Gini index, Human Development Index (HDI), Environmental Performance Index (EPI), a climate vulnerability index (ND-GAIN); see Tables S23-S26. As we now note in the “Additional National-Level Covariates” section of our Supplemental Materials, *“These variables could plausibly interact with cultural tightness, so we include them to account for other omitted variables that*

might otherwise obscure or drive the observed effects.” Since some of these national indices were significantly correlated (see Table S40), we tested models with these additional covariates in separate supplemental models.

Overall these results show that our primary analyses are robust to the inclusion of these additional covariates. The only difference we found is in the case of one of our four outcomes (tree planting), for one of three interventions (Pluralistic Ignorance): while not significant in our original analysis, the interaction between Pluralistic Ignorance intervention and cultural tightness reached statistical significance when controlling for the new national-level measures. However the effect was not in the predicted direction, such that the intervention group showed less conformity in culturally tight contexts. We discuss these results in the “Robustness Analyses” section of the Results.

“Furthermore, when controlling for additional country-level variables (i.e., the Environmental Performance Index [96], the Vulnerability Index of Notre Dame Global Adaptation Initiative Country Index [97], the Gini Index [98], and the Human Development Index [99]; see Tables S23-S26), the results remain largely consistent with the main models reported in Table 2. The only difference concerns tree planting: in the primary analysis, the interaction between cultural tightness and the Pluralistic Ignorance intervention was significant only in the model without covariates, whereas in the supplementary analyses it is significant when we control for additional national-level covariates (see Table S26).”

Other points:

P9, L198–206: The interaction effects for Pluralistic Ignorance are highlighted but not robust to corrections. The discussion should explicitly caution against overinterpreting these patterns without replication.

We have amended the Discussion section to explicitly caution readers against overinterpretation, as the few significant effects are the exception to our otherwise null findings and are not robust to corrections for family-wise error.

“However, these patterns should be interpreted with caution as these significant interactions are the exception and are not robust to family-wise error corrections.”

P17, L335–342: The speculation about climate change being a "unique domain" needs empirical support. Cite studies showing how norm interventions perform differently in climate vs. other collective action problems (e.g., public health).

In the final paragraph of our new limitations section, we now make a case for why the climate domain may function differently in the context of social norm interventions.

“Finally, this study is focused on the domain of climate change. This domain may be unique for several reasons. Climate change is a global collective action problem that is highly polarized and marked by an uneven distribution of both impacts and responsibility for its causes [7]. Addressing climate change is often thought to require responses from economically powerful nations [115-116] and individual action from wealthy, high-

emitting individuals. Given these features, climate change may present a particularly difficult test case for the generalizability of norm interventions across cultures—especially insofar as cultural contexts also differ in their histories of industrialization and emissions. At the same time, climate change is a complex problem with many uncertainties that requires collective action to address, potentially rendering social norms an especially potent influence on behaviors and beliefs. Nevertheless, we found that historical emissions and wealth do not correlate with cultural tightness (see Table S40), nor do we observe clear and consistent evidence that these national-level measures influence the main interaction results between tightness and the interventions (see Tables S31-S39). Future replications of this work in domains that are more relevant to a local or national context (e.g., public health) may therefore yield different results than those presented here.”

We are not aware of any research directly comparing social norm interventions in climate and non-climate domains, so our claim that climate change may constitute a unique domain remains necessarily tentative. However there are some who point to similarities between other international issues (e.g., COVID-19) and climate change (Burrows, Abellera, & Markowitz, 2023; Whomsley, 2021; Eom et al., 2025), so there may be other domains with similar features that are more suitable for these experimental tests.

References:

- Burrows, B., Abellera, C., & Markowitz, E. M. (2023). COVID- 19 and climate change: the social- psychological roots of conflict and conflict interventions during global crises. *Wiley Interdisciplinary Reviews: Climate Change*, 14(5), e837.
- Whomsley, S. R. (2021). Five roles for psychologists in addressing climate change, and how they are informed by responses to the COVID-19 outbreak. *European Psychologist*.
- Eom, K., Cole, J. C., Dickert, S., Flores, A., Jiga-Boy, G. M., Kogut, T., ... & Van Boven, L. (2025). It's all connected: Collectivism, climate change, and COVID-19. *Acta Psychologica*, 258, 105200.

Methods (P23, L473): Clarify how "representative samples" were defined (e.g., quotas, random sampling) and whether this aligns with standard practices in cross-cultural research.

We now provide a clearer explanation of what we mean by “representative samples” in the Table S2 note. Levels of representativeness varied across countries: in some countries the data were not representative along any demographic dimensions (most likely due to limited funding), while in other countries data were representative along one or more dimensions. We report a summary of country-level representativeness in Table S2, and point readers to Table 1 of Vlasceanu et al., 2024 for further details.

While representative samples allow for more reliable cross-cultural comparisons (see e.g., Tam & Milfont, 2020), this is not always possible, especially for global studies where different research teams have different capacities. We believe that lack of representativeness in some countries may be less of a concern here for the following reasons: the study is

experimental with randomization within the country, representativeness is not correlated with tightness ($\rho = -.21$, $p = .18$), and we control for demographic covariates. That said, we share this concern with the Reviewer and report this limitation in the Discussion.

“Despite offering the first systematic experimental test of the relationship between cultural tightness and conformity to norm information, the present research has several limitations. First, although our sample is considerably larger and spans more countries than previous studies, it was constrained to the 42 countries included in Vlasceanu et al.’s [83] and Eriksson et al.’s [47] datasets. Some of these countries have relatively small sample sizes (see Table S2), and the ManyLabs dataset includes uneven sampling and heterogeneous representativeness across countries, reflecting differences in local data collection procedures and capacities. These factors should be considered when interpreting the results both within and across nations. While our dataset includes demographic quotas in many countries, not all countries’ samples are fully representative of their population, and these deviations may limit the utility of national-level measures of cultural tightness. We do not expect that differences in representativeness are a confound here as we find no correlation between tightness and representativeness ($\rho = -.21$, $p = .18$; see Representativeness Checks in the Supplemental Materials). Furthermore, representativeness is arguably less of a concern in our study because our design is experimental with randomization occurring within countries. However, to account for differences in representativeness, we subset the sample to countries with low- vs. high-representativeness (see Tables S27-S30) and find only one significant interaction: cultural tightness and the Work-Together Norm interact to predict increased belief in climate change in the less-representative sample.”

References:

- Tam, K. P., & Milfont, T. L. (2020). Towards cross-cultural environmental psychology: A state-of-the-art review and recommendations. *Journal of Environmental Psychology*, 71, 101474. <https://doi.org/10.1016/j.jenvp.2020.101474>
- Vlasceanu, M., Doell, K. C., Bak-Coleman, J. B., Todorova, B., Berkebile-Weinberg, M. M., Grayson, S. J., ... & Lutz, A. E. (2024). Addressing climate change with behavioral science: A global intervention tournament in 63 countries. *Science advances*, 10(6), eadj5778. <https://doi.org/10.1126/sciadv.adj5778>

These critiques target gaps where additional analyses or discussion could substantially strengthen the paper’s contributions.

Thank you for your suggestions!

Reviewer #2

This is a review of the manuscript entitled “Assessing the Role of Culture in Moderating Social Norm Interventions: A Global Experiment,” submitted to *Nature: Communications Psychology*. To begin, the motivation to understand cultural variance in the effectiveness of different norm-based interventions is laudable, and beginning with cultural tightness is reasonable. There are also several strengths to the paper, including the large sample size and large number of sample locations, as well as the strong experimental control toward the goal of examining replicability. However, I also have several concerns about the methods for the cross-cultural hypothesis testing in the manuscript and the conclusions drawn, which are the focus below.

Thank you. We appreciate your comments.

1. Moving beyond cultures of the Global North is a motivation for this paper, and is in some ways handled effectively (e.g., many cultures analyses; appreciating that the Global North disproportionately causes the global climate crisis, while the Global South disproportionately suffers the consequences of it). However, a deep consideration of what this means for the research is lacking. Henrich, Heine, and Norenzayan (2010) present analysis that informs how the goals of this project as articulated may be too mismatched from the methods. Specifically, Henrich et al. (2010) examine how the research methodologies themselves can be too far removed from cultural contexts of interest, and they underscore how movement away from US samples to contexts that are less Western, Educated, Industrialized, Rich, and Democratic (WEIRD) often requires significant adaptation of research methodologies. The manuscript Discussion section almost acknowledges this in describing normative influence in Asia and how tight cultures require more proximal norm information, but consideration of the global generalizability of the methods and assumptions about methodological fit for the samples require attention. I unpack this more below in points 2, 6, and 8.

We appreciate the reviewer’s point that while our study moves beyond cultures of the Global North, the manuscript needed deeper consideration of what this means for research, particularly regarding the fit between our methods and the cultural contexts of interest, as raised by Henrich and colleagues (2010). To address this, we added a new section titled “*Why did cultural tightness fail to moderate the efficacy of social norm interventions?*” in the Discussion where we (a) acknowledge potential cultural fit issues for the interventions tested, (b) note that all three interventions were developed and primarily validated in the Global North, (c) highlight that the interventions were effective in the U.S. subsample but not in the full cross-cultural sample, and (d) speculate on why each intervention might not have shown the expected moderation effect in diverse cultural contexts. We also now note that ManyLabs Megastudies, while valuable for systematic analyses of overall effects, may be less suited for identifying locally optimized interventions or analyses of cross-cultural differences and could result in spurious conclusions where interventions and survey instruments are not cross-culturally validated and online, impersonal survey-based experiments and measures may not be

customary. As such, we note that future work should pre-test and ensure that interventions are theoretically and practically suited to the cultural contexts included in the study, and consider how the approach of the research itself—for example, an embedded experiment in an online survey vs. an in-person lab or field experiment— might vary in its ecological validity across different cultural contexts.

“[...] all of the social norm interventions are effective in shifting at least some of our specified outcomes when limiting our sample to the United States (see Table S22), yet these effects do not generalize to the full cross-cultural sample. This suggests that the interventions may be particularly attuned to the United States and potentially other countries in the Global North context and less well-suited to countries in the Global South. Importantly, however, cultural tightness does not perfectly map onto distinctions between the Global North and South, so while this is a general concern for ManyLabs Megastudies, it does not represent a direct confound for the present study.

Some studies outside of the United States have found that dynamic norms, togetherness norms, and descriptive norm information can be effective means of shifting behaviors (e.g., [28,101-102]); however, these interventions may need to be culturally-attuned to each setting to be effective—including potentially delivered via different modalities beyond impersonal, light-touch, online surveys. This could suggest that ManyLabs Megastudies, while important for systematically testing interventions across cultural contexts, are not ideally suited to identify locally-optimized and culturally-attuned interventions, especially since many of the tested interventions have been primarily validated in the Global North and proposed by scholars working in those settings. In this study, local samples were mostly collected by local researchers. However, the intervention text was submitted by experts who were largely based in the Global North. To better ensure that interventions are applicable and comparable across cultural contexts, future studies should pre-test and validate intervention materials. Additionally, interventions could be more strategically designed to theorize and test cross-cultural differences in the underlying mechanisms that make some interventions effective in one setting but not in others.”

Subsequently, we speculate on why the interventions might not have shown the expected moderation effects, emphasizing how differences in the type of norm invoked (i.e., national vs. location, descriptive vs. injunctive, established vs. emerging) may interact with cultural tightness to shape their effectiveness (see our new section “*Why did cultural tightness fail to moderate the efficacy of social norm interventions?*” for the full discussion).

In addition, in our limitations section we now elaborate on the lack of cross-cultural intervention validation and the importance of considering how injunctive versus descriptive norm emphasis may interact with cultural tightness. Drawing on the paper by Henrich and colleagues (2010), we now emphasize that moving beyond WEIRD populations often requires adapting methods. However, at the same time, we would like to highlight that this study was administered online and is unlikely to be reaching populations without familiarity with computer based surveys and access to the internet. We connect this to unequal contributions to, and consequences of,

climate change, and recommend participatory approaches to ensure cultural fit and reduce perceived power asymmetries in future global collaborations.

“[...] as Henrich and colleagues [111] argue, moving beyond WEIRD populations often requires not just a change in sampling frame but also substantial adaptation of research methodologies to ensure they are ecologically valid in the context of interest. Although effort was made to adapt study materials in consultation with local researchers, the interventions still reflect design choices rooted in U.S.-based studies, which may have reduced their effectiveness in some cultures. Furthermore, because this study was administered online, it is unlikely to have reached populations without internet access and familiarity with computer-based surveys; thus, our sampling frame systematically excludes certain populations—limiting representativeness—while simultaneously bolstering survey construct validity. Future work could address this tradeoff by combining online methods with culturally-resonant approaches and efforts to reach underrepresented populations while preserving validity.”

2. It is not clear in the manuscript, supplemental materials, or original dataset publication precisely who the participants were and how translations were managed. While this matters less for the original publication that was not focused on potential cultural differences, to draw conclusions about cultural variance – particularly about how tightness leads to reduced normative influence – requires this was all done to the highest standards.

Thank you for pointing out that our manuscript previously did not include the translation procedure or sufficient information about sample characteristics. We now summarize key aspects of the translation procedure in our Methods section, and point readers to Doell et al. (2024)’s Methods section for additional information. As we now note, translation was handled by researchers from that country, and back translations were conducted to help ensure fidelity: *“The procedure was the same in all countries, including translations to local languages and back translations (for further details see [84]).”* Additionally, we add in a footnote that local research teams were responsible for translating materials and flagging culturally-inappropriate content: *“The ManyLabs project was designed with a decentralized model in which local research teams residing in each nation were responsible for overseeing recruitment, obtaining IRB approval through their institutions, translating materials into their local native language, and flagging any content that might be culturally inappropriate.”*

In addition to the information about participants reported in Table S3, which are broken down by each country, we also discuss sample representativeness in Table S2, noting the samples were not always representative. We now provide further information in both the manuscript and the Supplementary Materials, and direct readers to the original publication by Vlasceanu et al. (2024) for additional information regarding sample composition.

As noted in response to Reviewer 1, we also ran additional analyses to test whether differences in representativeness across countries may have introduced a confound in our analyses. First, we ran a correlation between cultural tightness and level of representativeness to ensure that representativeness and tightness did not co-vary. We find no significant correlation between these two measures ($\rho = -.21$, $p = .18$). Additionally, in subgroup analyses

where we ran our main analyses in low and high representativeness countries, we find that tightness significantly interacts with the Work-Together Norm intervention to predict Belief in Climate Change in the less representative sample, but not in the high representativeness sample (see Table S27). However, this was the only difference in significance between the two samples (see also Tables S28-S30). Thus, it does not appear that sample representativeness changes the hypothesized moderation effects of the norm interventions. We added a summary of our findings in our Limitations section:

“[...] to account for differences in representativeness, we subset the sample to countries with low- vs. high-representativeness (see Tables S27-S30) and find only one significant interaction: cultural tightness and the Work-Together Norm interact to predict increased belief in climate change in the less-representative sample.”

References:

- Doell, K. C., Todorova, B., Vlasceanu, M., Bak Coleman, J. B., Pronizius, E., Schumann, P., ... & Lutz, A. E. (2024). The International Climate Psychology Collaboration: Climate change-related data collected from 63 countries. *Scientific data*, 11(1), 1066. <https://doi.org/10.1038/s41597-024-03865-1>
- Vlasceanu, M., Doell, K. C., Bak-Coleman, J. B., Todorova, B., Berkebile-Weinberg, M. M., Grayson, S. J., ... & Lutz, A. E. (2024). Addressing climate change with behavioral science: A global intervention tournament in 63 countries. *Science advances*, 10(6), eadj5778. <https://doi.org/10.1126/sciadv.adj5778>

3. The experimental manipulations confound the type of normative influence with other factors, including the presence of quantitative data and explicit requests to take actions later measured in dependent variables. This makes it challenging to disentangle the effects of the intended normative influence relative to other confounded factors.

We agree that the three norm interventions examined vary in a variety of ways beyond the type of norm intervention used and thus that differences between them should not necessarily be attributed to the specific type of norm intervention. However, our primary goal in this study is not to contrast the norm interventions, but rather to assess whether cultural tightness moderates the general effectiveness of norm-based interventions. To clarify this in the manuscript, we have revised the introduction to emphasize that our question is about norm interventions overall.

“Nevertheless, while the three interventions vary in terms of the social norm information conveyed, and how it is communicated, the primary goal of this work is not to attribute differences in moderation to the specific norm interventions in comparison to each other. It is, instead, to identify whether cultural context—and specifically its tightness or looseness—moderates the influence of norm information on attitudes and behaviors relative to a neutral control.”

The variation in norm interventions allows us to examine generalizability across three common norm interventions. We have added this to the introduction section:

“Together, these interventions reflect a diversity of norm manipulations used in the literature, allowing us to generalize across specific norm intervention design choices.”

4. The numbers represented in the quantitative data for normative and pluralistic ignorance manipulations differed for each country. This is important for normative influence, but as the norm setting data shift as we move away from the US, the influence patterns should too – moving toward the norm will be different when the norm is different.

We appreciate the reviewer’s observation that norm-setting data vary across countries and that this variation may have implications for normative influence. We agree that moving toward a different norm might yield different behavioral effects. We now address this issue more explicitly in our new section “Robustness analyses” across three supplemental analyses conducted in the Pluralistic Ignorance condition.

Our first analysis includes an “accuracy” score for each participant, defined as the difference between their estimate of their national norm and the actual norm. Although accuracy was significantly associated with each outcome, including this variable did not alter our interaction effects (Table S13). Our second analysis uses a binary indicator distinguishing participants who over- vs. under-estimated the norm. Because the intervention is designed to correct systematic underestimation, overestimation could plausibly dampen the effects or even backfire. However, including this indicator did not change our results (Table S14). Finally, we included a new analysis controlling for the specific norm percentage displayed to participants. This variable significantly, but modestly, predicted willingness to post on social media (an effect not robust to family-wise error correction), and was non-significant in the other three models (Table S15). Together, these analyses suggest that our core results are robust to variation in norm estimates and the normative information provided to participants.

We include two paragraphs summarizing these effects in the “Robustness analyses” section of our Results:

“In the Pluralistic Ignorance condition, all participants were asked to give an estimate of their country’s norm prior to being informed about that norm. Participants in the Control condition were also asked to provide this estimate. While the exact percentage did vary across countries, it is important to note that the manipulation always described a majority view in that nation, with nearly all nations reporting a super-majority (i.e., more than 60% of the population). Nonetheless, because the actual norm varies by country, participants’ estimates reflect differing distances from their national baseline. This is critical for normative influence: both the distance and direction of participants’ misperceptions could introduce individual heterogeneity in the effect of the norm intervention—to the extent that the heterogeneity in accuracy varies across cultural context, this could impact our main inferences. To account for this, we calculated an “accuracy” score for each participant by calculating the difference between their estimate and the actual norm in their country/region. This difference score captures how close an individual is to the country-specific norm at baseline. This score was significantly associated with each outcome variable, indicating the importance of prior norm perceptions, but accounting for baseline estimates did not change the interaction effects (see Table S13), suggesting

that our core results are robust to cross-national variation in norm estimates and the data provided by the intervention.

We also conducted two additional robustness analyses in the Pluralistic Ignorance condition. First, we ran these analyses using a binary indicator to distinguish respondents who over- or under-estimated the norm, since this may impact the efficacy of the Pluralistic Ignorance intervention. Specifically, the intervention was designed to correct systematic underestimation of existing social norms; therefore, for participants who overestimate the norm, learning that the norm is actually lower than they expected may actually have the opposite effect on the dependent variables. Including this indicator did not change our results (see Table S14). Second, we ran these analyses controlling for the specific norm percentage displayed to participants in the Pluralistic Ignorance condition. For all outcomes, the inclusion of this covariate did not change our primary findings. In fact, while norm percentage was a significant predictor of willingness to post on social media, this effect was small and it was unrelated to all other outcomes (see Table S15)."

5. The manuscript also acknowledges how the dynamic norm and pluralistic ignorance experimental conditions focused on the most proximal data available (e.g., Kenya would be included for Kenyan participants), but it is unclear which comparison countries were included in each country's stimulus materials. Given the disproportionate responsibility and consequences for different countries, the comparison groups could strongly influence participants in many ways.

Researchers were instructed to pipe in their country in the first sentence of the dynamic norm statement so that the intervention emphasized a dynamic norm in the participant's country first. Next, participants saw a standardized figure showing trends from the same set of ten countries (see Figure 1a) as well as the global average. This graphic was designed to illustrate that the norm was changing worldwide. Thus, all participants saw the same set of global comparisons, so there is no separate analysis for countries that may or may not have appeared in the figure.

However, we acknowledge that the comparison set could carry different meanings based on a country's current and historical circumstances. In particular, if participants saw their country included in the figure, and observed that their country was increasing in mitigative action or concern while historical emitters showed less of an increase, this might lead participant in that country to believe that their country should not take additional action. For example, participants in Kenya may have noticed that their country showed higher concern than the UK, Canada, Australia, or the US. To address this possibility, our main models control for national-level differences in GDP and CO₂ emissions per capita—both reasonable proxies for a country's historical and economic contribution to global emissions (Herzog, Pershing, & Baumert, 2005; Pauw et al., 2014). Additionally, we found that GDP per capita and CO₂ emissions per capita do not correlate with cultural tightness (ρ s > .3, see Table S40), and are therefore unlikely to confound our key inferences.

Additionally, we conducted robustness analyses (see Tables S23-26 in the Supplemental Materials) in which we control for the Environmental Performance Index (EPI), a climate vulnerability index (ND-GAIN), as well as the Gini index, and the Human Development Index (HDI). Since bivariate correlations between the national-level variables demonstrate that some of these variables correlate (see Table S40), we proceeded by adding each additional variable individually to a new model (see Tables S23-S26). Overall, these additional checks provide a more comprehensive measure of national environmental responsibility. We found that our primary findings do not change: cultural context does not moderate the effect of the Dynamic Norm intervention on any of the four outcomes.

Finally, to test whether the interventions in historically high- and low-emitting countries may have different outcomes, we re-ran the main models dividing the sample into countries with low vs. high CO₂ cumulative emissions (see Tables S31-S34 in the Supplemental Materials). We found that the results for the Dynamic Norm do not change. We also ran a model testing a three-way interaction between cultural tightness, the comparison between the control and the Dynamic Norm intervention, and cumulative emissions. We do not find significant moderation by historical emissions for any of the outcomes (see Table S39). We have added a sentence to the discussion to acknowledge these new analyses: *“However, we found that historical emissions and wealth do not correlate with cultural tightness (see Table S40), and we do not see any clear and consistent evidence that these national-level measures influence the main interaction results between tightness and the interventions (see Tables S31-S39).”*

Together, we hope these controls can help account for the kind of structural disparities that might influence how participants interpret norm information across countries. While no statistical control can fully substitute for experimental manipulation, we believe these variables serve as meaningful proxies and provide sufficient reassurance that our results are not simply an artifact of cross-national comparison effects in the Dynamic Norm condition. Additionally, the comparison to other countries is not present for either the Work Together or Pluralistic Ignorance conditions, where we find very similar results.

References:

Herzog, T., Pershing, J., & Baumert, K. A. (2005). *Navigating the numbers*.

http://pdf.wri.org/navigating_numbers.pdf

Pauw, P., Bauer, S., Richerzhagen, C., Brandi, C., & Schmole, H. (2014). Different perspectives on differentiated responsibilities. *A State-of-the-Art Review of the Notion of Common but Differentiated Responsibilities*. German Development Institute, Bonn. https://www.idos-research.de/uploads/media/DP_6.2014..pdf

6. Further, it is unclear if the cultural variance or heterogeneity between (and within) samples is due to differences in tightness or to methodological artifacts or to histories of colonization, which also relates to the quality of translations and the presentation of the research to participants. If the study materials read as if created by foreigners (which would be very reasonable given that they were for most countries), this can be experienced with a shadow of colonial history by participants, especially in the most economically and environmentally disadvantaged countries. Many of these countries have been most exploited by formal colonization and business and governmental

extraction by the Global North. The resistance to influence and even reactance could be based on local efforts to reclaim control, which would require emic considerations of how these interventions might be experienced in the cultural contexts coded as tight in this research. Even assuming that materials were translated brilliantly, information about foreign researcher leads who are Western or more economically powerful within continents could still trigger such resistance.

We appreciate the reviewer's thoughtful reflections on the broader sociopolitical context in which cross-cultural research is conducted. We agree that the potential for interventions to be perceived through a colonial lens, particularly in countries with histories of exploitation, raises critical issues for the design, implementation, and interpretation of global research.

To help address this, the ManyLabs project was designed with a decentralized model in which local research teams who resided in each nation were responsible for overseeing recruitment, obtaining IRB approval through their institutions, translating material into their local native language, and flagging any content that might be culturally inappropriate. Additionally, participants were recruited by researchers from their own country, and institutional ethics approval referenced "international collaboration" without specifying that the research was led by researchers in the United States. There was no mention of New York University in the participant-facing materials. These steps were taken to ensure that the research would not appear to be directly imposed by an external entity, and are now described in detail in a footnote in the discussion.

That said, we fully acknowledge that the perception of researchers (even those from within one's own country) could be perceived as aligned with the Global North despite these careful design decisions, especially given the unequal global distribution of research infrastructure. We believe this is a pervasive challenge for international science.

We also note that our analyses focus on cultural tightness, which is conceptually and empirically distinct from Global North vs. South distinctions. Tight cultures include countries from both the Global North and South. Nonetheless, the reviewer's broader point stands: resistance to norm-based influence could reflect post-colonial power dynamics. This is important for cross-cultural research to address and carefully consider. We now include the following paragraph in the Discussion:

"Third, although local researchers adapted study materials and recruited participants, it is possible that participants in some countries may have perceived the interventions as externally imposed. Even decentralized collaborations like this one may be interpreted through a colonial lens, particularly in historically exploited or economically disadvantaged settings. While this may influence the interpretation of and response to the materials, particularly in Global South countries, cultural tightness is distinct from whether a country is in the Global North or South and so we do not expect this influence to alter the results presented here. Nonetheless, future cross-cultural research should make efforts to ensure consistent sampling, the use of culturally-resonant and validated materials, and participatory approaches to intervention and study design. These efforts can reduce the likelihood of potential confounds that impact the inferences that can be drawn about cultural differences. Additionally, while it is not possible to experimentally manipulate national levels of cultural tightness, future research might attempt to replicate

the theorized interaction using pseudo-experimental designs to examine differences in conformity to norm interventions between people in the same cultural context who come from tight or loose backgrounds [66]. Our null results for national levels of cultural tightness do not imply that individual-level tightness would also yield null results, and testing this possibility directly remains an important next step for future research.”

7. The focus on environmental sustainability introduces more concerns about the methodological fit for participants. For example, while gross domestic product per capita and income were covariates, neither were tested as interacting with the interventions. These analyses are both important for the contribution relative to disproportionate environmental crisis responsibility and consequences as articulated in the manuscript.

We agree that sustainability interventions in high and low emitting countries may yield different outcomes, and that these differences may be related to cultural tightness. Likewise, higher GDP per capita implies a greater national capacity to respond to environmental crises. To examine whether these factors might confound our analyses, we first tested their associations with our main independent variable, cultural tightness. We found no correlations between cultural tightness and either GDP per capita or CO₂ emissions (see Table S40). Thus, although GDP per capita and emissions might influence outcomes (and are controlled for in our main analyses), they are unlikely to confound the core relationship between cultural tightness and intervention effects.

To further address the reviewer’s concern, we provide two methods to further assess the possibility that countries with different emissions or GDP per capita might show differential effects for the moderation of norm interventions by cultural tightness. Specifically, we re-ran the main model, dividing the sample into countries with low vs. high cumulative CO₂ emissions (see Tables S31-S34 in the Supplementary Materials) and low- and high-GDP per capita (see Tables S35-S38 in the Supplementary Materials).

Results show scattered and inconsistent interactions in the Pluralistic Ignorance condition, and no effects in the other two conditions. Specifically, we find that the Pluralistic Ignorance intervention interacts with cultural tightness to yield significantly lower belief in climate change in countries with *low* emissions (see Table S31) and *low* GDP per capita (see Table S35). Conversely, for the tree planting task, we find that this interaction is significant only in countries with *high* emissions (see Table S34), and that for policy support, the intervention interacts with tightness in countries with *high* GDP per capita (see Table S36). However, all other interaction effects are non significant. We now summarize these findings in the Discussion section: *“However, we found that historical emissions and wealth do not correlate with cultural tightness (see Table S40), and we do not see any clear and consistent evidence that these national-level measures influence the main interaction results between tightness and the interventions (see Tables S31-S39).”*

Finally, per the reviewer’s suggestion, we examined whether GDP per capita, personal income, or emissions moderate the impact of the interventions. Below, we report the results of the models testing the interaction between the interventions and each variable, while controlling for individual-level covariates. We find that GDP interacts with the Dynamic Norm and Work-Together Norm to predict belief in climate change (see Table A), the Pluralistic Ignorance Norm

interacts with CO₂ emissions to predict belief in climate change (see Table B), and that personal income interacts with Pluralistic Ignorance Norm to predict tree planting (see Table C). However, these significant effects are the exception, as all other interactions are not significant. When taken together, these analyses provide limited evidence that GDP per capita, emissions, or income systematically influence our results.

Table A. Predicting climate change attitudes and behavior with the 2-way interaction between Norm interventions and GDP per capita, while controlling for the individual-level covariates.

Predictors	Belief in Climate Change	Policy Support	Social Media Post	Tree Planting
	Estimates	Estimates	Estimates	Estimates
Intercept	77.89 *** (75.72 – 80.06)	70.05 *** (68.17 – 71.93)	0.54 *** (0.47 – 0.61)	4.23 *** (3.97 – 4.49)
Dynamic Norm	1.09 (-0.05 – 2.23)	1.38 ** (0.45 – 2.32)	0.07 *** (0.04 – 0.10)	-0.08 (-0.24 – 0.08)
Work-together Norm	-0.11 (-1.26 – 1.03)	0.34 (-0.60 – 1.28)	0.05 *** (0.03 – 0.08)	-0.37 *** (-0.53 – -0.21)
Pluralistic Ignorance	0.93 (-0.21 – 2.07)	0.77 (-0.17 – 1.70)	0.02 (-0.01 – 0.04)	-0.13 (-0.29 – 0.03)
GDP per capita	-0.10 (-1.23 – 1.03)	-0.82 (-1.80 – 0.17)	-0.05 ** (-0.08 – -0.01)	-0.01 (-0.14 – 0.13)
Age	-0.09 *** (-0.12 – -0.06)	-0.00 (-0.03 – 0.02)	-0.00 *** (-0.00 – -0.00)	0.04 *** (0.03 – 0.04)
Gender: Female	4.11 *** (3.28 – 4.93)	1.49 *** (0.81 – 2.17)	-0.05 *** (-0.07 – -0.03)	0.47 *** (0.35 – 0.58)
Gender: Non-binary/other	4.48 (-0.70 – 9.66)	3.62 (-0.65 – 7.89)	-0.10 (-0.22 – 0.01)	0.39 (-0.34 – 1.13)
Education	0.86 ** (0.21 – 1.51)	1.42 *** (0.89 – 1.96)	0.04 *** (0.03 – 0.06)	0.06 (-0.03 – 0.15)
Income	0.40 ** (0.15 – 0.64)	0.39 *** (0.18 – 0.59)	0.00 (-0.00 – 0.01)	0.02 (-0.02 – 0.05)
Political orientation	-0.22 *** (-0.24 – -0.20)	-0.14 *** (-0.15 – -0.13)	0.00 *** (0.00 – 0.00)	-0.01 *** (-0.01 – -0.01)
Dynamic Norm * GDP per capita	-0.44 (-1.02 – 0.13)	-0.29 (-0.76 – 0.18)	-0.00 (-0.02 – 0.01)	-0.03 (-0.11 – 0.05)
Work-together Norm * GDP per capita	-1.00 *** (-1.57 – -0.42)	-0.42 (-0.89 – 0.05)	0.00 (-0.01 – 0.01)	0.02 (-0.06 – 0.10)
Pluralistic Ignorance * GDP per capita	-0.88 ** (-1.45 – -0.31)	-0.46 (-0.93 – 0.00)	-0.01 (-0.02 – 0.00)	0.02 (-0.06 – 0.10)
Random Effects				
σ^2	521.38	349.53	0.20	10.40
T ₀₀	37.47 Country	28.90 Country	0.04 Country	0.47 Country
ICC	0.07	0.08	0.16	0.04
N	39 Country	39 Country	39 Country	39 Country
Observations	12474	12436	9537	12483
Marginal R ² / Conditional R ²	0.069 / 0.131	0.048 / 0.120	0.054 / 0.209	0.041 / 0.083

* $p < 0.05$ ** $p < 0.01$ *** $p < 0.001$

Table B. Predicting climate change attitudes and behavior with the 2-way interaction between norm interventions and cumulative CO₂ emissions, while controlling for the individual-level covariates.

Predictors	Belief in Climate Change	Policy Support	Social Media Post	Tree Planting
	Estimates	Estimates	Estimates	Estimates
Intercept	78.06 *** (75.92 – 80.20)	70.02 *** (68.07 – 71.97)	0.53 *** (0.46 – 0.60)	4.24 *** (3.98 – 4.50)
Dynamic Norm	0.76 (-0.50 – 2.03)	1.02 (-0.01 – 2.05)	0.07 *** (0.04 – 0.10)	-0.13 (-0.31 – 0.05)
Work-together Norm	-0.52 (-1.79 – 0.74)	0.04 (-1.00 – 1.07)	0.05 *** (0.02 – 0.08)	-0.38 *** (-0.56 – -0.20)
Pluralistic Ignorance	0.35 (-0.91 – 1.60)	0.52 (-0.51 – 1.55)	0.01 (-0.02 – 0.04)	-0.10 (-0.28 – 0.08)
Cumulative CO ₂ Emissions	-2.41 (-4.82 – 0.00)	-1.77 (-4.01 – 0.47)	-0.00 (-0.09 – 0.09)	-0.03 (-0.31 – 0.25)
Age	-0.09 *** (-0.12 – -0.06)	-0.00 (-0.03 – 0.02)	-0.00 *** (-0.00 – -0.00)	0.04 *** (0.03 – 0.04)
Gender: Female	4.14 *** (3.32 – 4.97)	1.50 *** (0.82 – 2.18)	-0.05 *** (-0.07 – -0.03)	0.46 *** (0.35 – 0.58)
Gender: Non-binary/other	4.32 (-0.86 – 9.50)	3.55 (-0.72 – 7.82)	-0.10 (-0.22 – 0.01)	0.39 (-0.34 – 1.12)
Education	0.88 ** (0.23 – 1.53)	1.44 *** (0.90 – 1.97)	0.04 *** (0.03 – 0.06)	0.06 (-0.03 – 0.15)
Income	0.40 ** (0.16 – 0.65)	0.39 *** (0.19 – 0.59)	0.00 (-0.00 – 0.01)	0.02 (-0.02 – 0.05)
Political orientation	-0.22 *** (-0.24 – -0.20)	-0.14 *** (-0.15 – -0.13)	0.00 *** (0.00 – 0.00)	-0.01 *** (-0.01 – -0.01)
Dynamic Norm * Cumulative CO ₂ Emissions	0.40 (-0.29 – 1.08)	0.46 (-0.11 – 1.02)	-0.00 (-0.02 – 0.01)	0.07 (-0.03 – 0.16)
Work-together Norm * Cumulative CO ₂ Emissions	0.50 (-0.19 – 1.19)	0.38 (-0.18 – 0.95)	0.00 (-0.01 – 0.02)	0.01 (-0.09 – 0.11)
Pluralistic Ignorance * Cumulative CO ₂ Emissions	0.73 * (0.04 – 1.41)	0.32 (-0.24 – 0.89)	0.01 (-0.01 – 0.02)	-0.04 (-0.14 – 0.06)
Random Effects				
σ ²	521.82	349.59	0.20	10.40
T ₀₀	35.82 Country	31.49 Country	0.05 Country	0.47 Country
ICC	0.06	0.08	0.19	0.04
N	39 Country	39 Country	39 Country	39 Country
Observations	12474	12436	9537	12483
Marginal R ² / Conditional R ²	0.087 / 0.146	0.051 / 0.129	0.013 / 0.205	0.041 / 0.082

* $p < 0.05$ ** $p < 0.01$ *** $p < 0.001$

Table C. Predicting climate change attitudes and behavior with the 2-way interaction between norm interventions and personal income, while controlling for the other individual-level covariates.

	Belief in Climate Change	Policy Support	Social Media Post	Tree Planting
Predictors	Estimates	Estimates	Estimates	Estimates
Intercept	77.72 *** (75.57 – 79.88)	69.78 *** (67.83 – 71.72)	0.53 *** (0.46 – 0.60)	4.23 *** (3.97 – 4.48)
Dynamic Norm	1.08 (-0.06 – 2.22)	1.38 ** (0.44 – 2.31)	0.07 *** (0.04 – 0.10)	-0.07 (-0.24 – 0.09)
Work-together Norm	-0.16 (-1.30 – 0.99)	0.33 (-0.61 – 1.27)	0.05 *** (0.03 – 0.08)	-0.36 *** (-0.52 – -0.20)
Pluralistic Ignorance	0.93 (-0.21 – 2.07)	0.76 (-0.17 – 1.70)	0.02 (-0.01 – 0.04)	-0.12 (-0.28 – 0.04)
Income	0.61 ** (0.17 – 1.05)	0.45 * (0.09 – 0.81)	-0.00 (-0.01 – 0.01)	0.07 * (0.01 – 0.13)
Age	-0.09 *** (-0.12 – -0.06)	-0.00 (-0.03 – 0.02)	-0.00 *** (-0.00 – -0.00)	0.04 *** (0.03 – 0.04)
Gender: Female	4.13 *** (3.30 – 4.95)	1.50 *** (0.83 – 2.18)	-0.05 *** (-0.06 – -0.03)	0.46 *** (0.34 – 0.58)
Gender: Non-binary/other	4.20 (-0.98 – 9.38)	3.53 (-0.74 – 7.80)	-0.10 (-0.22 – 0.01)	0.39 (-0.34 – 1.12)
Education	0.87 ** (0.22 – 1.52)	1.43 *** (0.89 – 1.96)	0.04 *** (0.03 – 0.06)	0.06 (-0.03 – 0.15)
Political orientation	-0.22 *** (-0.24 – -0.20)	-0.14 *** (-0.15 – -0.13)	0.00 *** (0.00 – 0.00)	-0.01 *** (-0.01 – -0.01)
Dynamic Norm *	-0.43 (-1.02 – 0.16)	-0.10 (-0.58 – 0.38)	0.01 (-0.01 – 0.02)	-0.04 (-0.12 – 0.04)
Work-together Norm *	0.10 (-0.49 – 0.69)	-0.11 (-0.59 – 0.38)	0.00 (-0.01 – 0.02)	-0.07 (-0.16 – 0.01)
Pluralistic Ignorance *	-0.54 (-1.13 – 0.06)	-0.05 (-0.54 – 0.44)	0.01 (-0.00 – 0.02)	-0.10 * (-0.18 – -0.01)
Random Effects				
σ^2	521.72	349.67	0.20	10.40
T ₀₀	37.72 Country	32.06 Country	0.05 Country	0.46 Country
ICC	0.07	0.08	0.19	0.04
N	39 Country	39 Country	39 Country	39 Country
Observations	12474	12436	9537	12483
Marginal R ² / Conditional R ²	0.066 / 0.129	0.035 / 0.117	0.013 / 0.200	0.041 / 0.082

* $p < 0.05$ ** $p < 0.01$ *** $p < 0.001$

8. As a smaller point, the computer numeracy work for planting trees likely feels much stranger to participants who are extremely economically disadvantaged and in less industrialized and economically developed contexts.

We appreciate the Reviewer's feedback on the tree-planting task (the WEPT), and have added this point to the Discussion (see below). The WEPT was chosen as a primary outcome in the original dataset because it is meant to be an incentivized and costly proenvironmental behavior that does not require monetary contributions. In this way, it moves beyond intentions yet does not introduce financial decisions that are likely to be approached very differently depending on economic circumstance. However, it is indeed likely that time effort is also related to economic advantage, and that spending time on this is more possible for some than others—and perhaps less unusual as well. Additionally, country differences in contribution to emissions may change individual perceptions about whether they should in fact engage in this behavior. Finally, critics argue that this measure inadequately generalizes to real-world environmental behavior. We now include these points in the Discussion.

“Our only behavioral outcome, the tree planting task (the WEPT), showed backfire effects across most interventions. This may reflect concerns that the tree-planting task may not generalize to real-world, high-impact environmental behaviors. Indeed, while some scholars argue that the WEPT is an effective measure of pro-environmental behavior [112-113], others have found that it only weakly relates to actual carbon footprints, suggesting that this task may not capture meaningful environmental impact [114]. Furthermore, although the WEPT is a costly measure of behavior that requires participants to take time and effort to plant more trees, thereby avoiding explicit monetary decisions, its use across countries with starkly different economic advantages may still raise concerns about comparability when time has opportunity costs and these may be different across country contexts and individuals. To the best of our knowledge, the WEPT had been validated only in a handful of countries (e.g., in Belgium, [87]; in the USA, and South Africa [113], and in the UK, [113-114]), so it remains unclear to what extent economic background may influence performance across cultures. Additionally, it is possible that the task itself is unfamiliar and unintuitive for participants in many contexts.”

In closing, again, scale of data collection and importance of the research questions deserve recognition. I hope my suggestions help the authors to develop the paper to its potential, whether eventually published at Nature Communications Psychology or elsewhere.

Thank you for your insightful and detailed comments.

Reviewer #3

This work examines the role of cultural tightness in moderating the effectiveness of social norm interventions aimed at promoting climate change attitudes and behaviors. The topic is very interesting. Detailed comments are provided below.

Thank you for your comments.

1. Theoretical Framework. The authors present a reasonably comprehensive overview of the literature on cultural tightness and looseness. However, the rationale for including the specific set of social norm persuasion strategies – the pluralistic ignorance norm message, the dynamic norm message, and the Work Together Norm message – requires further clarification. What are the key distinctions between these three types of persuasive information, and what are the a priori hypothesized relationships between them in the context of cultural tightness? The study would benefit from a clearer articulation of the unique mechanisms by which each norm message is expected to influence individuals in cultures with varying degrees of tightness.

We agree it is important to clarify both the rationale for including these specific norm interventions and the theoretical scope of our research questions. As you suggest, the interventions differ in meaningful ways, such as in the types of norms they employ (descriptive vs. injunctive, dynamic vs. static) and their referent groups, among other differences. We have clarified our language in the Introduction when introducing the interventions.

“The interventions considered here have primarily been shown to be effective at shifting behaviors in the Global North, as such it is possible that they could have heterogeneous effects across cultural contexts given that they differ in terms of the norm referent group (e.g., neighborhood, nation, global region, the world), the type of social norm highlighted (e.g. injunctive, descriptive, dynamic), and which of the outcome variables they mention—we return to this possibility in the discussion.”

However, we also make it clear that our primary theoretical question does not primarily focus on the differences between the interventions themselves, but rather whether the effectiveness of norm interventions in general is moderated by cultural tightness. We thus see the different norm interventions as an opportunity to see whether this hypothesis generalizes across different types of norm interventions, acknowledging that some of the differences are likely to interact with cultural tightness in nuanced ways. We sharpened our language to more explicitly state this in the Introduction.

“Nevertheless, while the three interventions vary in terms of the social norm information conveyed, and how it is communicated, the primary goal of this work is not to attribute differences in moderation to the specific norm interventions in comparison to each other. It is, instead, to identify whether cultural context—and specifically its tightness or looseness—moderates the influence of norm information on attitudes and behaviors relative to a neutral control. Thus, our primary hypothesis is that the strength of different social norm interventions will be similarly moderated by cultural tightness. In exploratory

analyses, we examine their effects individually and use these findings to suggest avenues for future work.”

We return to these distinctions in the Discussion section when interpreting the exploratory comparisons between interventions. There, we speculate about how specific features of each intervention may interact with cultural context, while reiterating the limitations of such comparisons.

“Different social norm interventions are linked to distinct psychological mechanisms, which may vary in relevance or effectiveness depending on the cultural context. For instance, the Work-Together Norm intervention—which emphasizes an established group norm, highlights collective action, and invites individuals to ‘join in’—had a stronger effect on climate beliefs in tighter cultures. This may be because it more clearly communicates social expectations through injunctive, rather than descriptive, language (e.g., “We need to reduce our carbon footprint”) and offers direct behavioral guidance aligned with the outcome measures (e.g., “Donate to tree planting organizations”). Theories of cultural tightness pertain primarily to injunctive norms (i.e., explicit messages of what others should or should not do) [43-45]. Accordingly, the stronger effects of the Work-Together Norm in tight cultures may reflect its alignment with cultural sensitivity to prescriptive and proscriptive expectations.

In contrast, conformity to Dynamic Norm interventions requires individuals to align with emerging trends—attitudes or behaviors that are not yet established norms, but may become so in the future. In tighter cultures, where adherence to existing norms is stronger, people may be slower to adopt emerging trends, or may even react negatively if they are perceived as conflicting with existing norms. Similarly, Pluralistic Ignorance interventions may be more effective in contexts where people are less aware of prevailing social norms. This may be more common in looser cultures, where behaviors and beliefs vary more, and where norm awareness matters less due to weaker social consequences for deviation [41]. In line with this, forthcoming work suggests that pluralistic ignorance about willingness to combat climate change is more pronounced in loose, as compared to tight, cultures [106].”

In short, while the primary theoretical aim of the paper is to test whether cultural tightness moderates the effect of norm interventions in general, in the Discussion we explore how the distinct mechanisms for each intervention might impact our pattern of results. We hope this revision clarifies the theoretical scope of the study.

Furthermore, the justification for focusing on climate belief (H1), support for mitigation policies (H2), intentions to share climate change information on social media (H3), and effortful mitigation behavior (H4) is not sufficiently substantiated. Why were these specific outcome variables selected, and what theoretical rationale connects them to the social norm interventions and cultural tightness? A more robust theoretical foundation, supported by existing literature, is needed to explain the choice of these outcomes. Are these outcomes intended to capture attitude change, behavioral intention, actual

behavior, or all three? Clarifying the relationships among these different levels of outcomes would strengthen the theoretical framework.

Thank you for this suggestion. We have now added a paragraph in the Introduction to explain the role of each outcome. Specifically, we now explicitly articulate that each of the four outcomes was selected to capture distinct, complementary facets of climate action, spanning beliefs (a precursor of climate action), policy support (structural-level attitudinal commitment), willingness to publicly endorse climate action (a measure of the intention to influence peers), and an effortful and incentivized behavior. We state this in the following paragraph:

“These outcomes are important, in different ways, for broad-scale climate action. Belief in climate change is a well-established precursor to action—people are unlikely to change their behavior if they do not believe climate change is real or human-caused [77-78]. Since policy support is critical for achieving structural change [79], and public support for climate policies strongly predicts whether a policy is passed into law [80], measuring policy support captures an important attitudinal precursor to systemic change. Willingness to publicly express support for climate action relates to social influence and the diffusion of pro-climate norms, which are central to processes of collective action [73], yet has the limitation of being a behavioral intention. The inclusion of the WEPT addresses this limitation by including a costly, incentivized behavioral measure [81] and expands the study to also include individual mitigation actions [82]. The norm interventions tested here provide information that can be used to update second-order beliefs (i.e., what people believe others think and do), and were thus designed to shift individual attitudes and behaviors via conformity to norms.”

This addition clarifies how the selected outcomes map onto different levels of engagement (beliefs, intentions, and behaviors) and their relation to climate mitigation. We hope this revision addresses the reviewer’s concern by providing a clearer theoretical foundation for our analytic framework.

2. Methodology. The methodology section presents a design involving three different social norm interventions, each underpinned by distinct psychological mechanisms. The materials used in these interventions appear to differ substantially in both content and structure. A more detailed description and justification of these differences is needed. However, the control condition seems rudimentary, consisting of "reading a non-related text from Great Expectations by Charles Dickens." What specific construct(s) was this control condition designed to control for?

The aim of the control condition was to have it be similar to the other conditions (e.g., requiring people to read text) in terms of time and effort required but unrelated to climate change (for more details, see Doell et al., 2024). In our procedure section, we now clarify that the control condition was designed to be comparable to the treatment conditions in terms of time and effort required without relating to climate change.

“Participants in the control condition completed a task that was matched in terms of time and effort to the treatment conditions (i.e., they were asked to read an excerpt from

Great Expectations by Charles Dickens that was unrelated to climate change) before completing all questions used to assess the dependent variables.”

The interventions were submitted by experts and were evaluated by an independent panel as part of the megastudy procedure (see Vlasceanu et al., 2024 for information on the selection procedure). Thus, the specific design decisions were informed both by what had been shown to be effective in prior studies and by the evaluation process itself. This procedure naturally introduces differences not only in the type of normative content conveyed, but also in other design features across the interventions that could be consequential for their effectiveness. We have revised our Introduction section to discuss differences between the interventions, and explain how such variability helps provide generalizability for the hypothesized moderation.

“In the present research we test three norm interventions that vary on the above dimensions. The Pluralistic Ignorance message asks participants to estimate the percentage of people in their country who believe that climate change is a global emergency—implicitly invoking static descriptive and prescriptive injunctive norms. It then reveals the actual prevalence of people in the participant’s country who believe climate change is a global emergency, thereby correcting systematic underestimation of existing social norms [70-72]. The Dynamic Norm message describes recent increases in public concern about climate change, support for climate policies, and approval of sustainable behaviors, thus conveying both dynamic prescriptive injunctive norms and descriptive norms [29,33,73]. Finally, the Work Together Norm message communicates strategies to reduce one’s carbon footprint, while emphasizing the prevalence of such behaviors, and combines both prescriptive and proscriptive injunctive norm information with descriptive norm information [69,74]. Together, these interventions reflect a diversity of norm manipulations used in the literature, allowing us to generalize across specific norm intervention design choices.”

Finally, we return to this point in our new discussion section titled “*Why did cultural tightness fail to moderate the efficacy of social norm interventions?*” to consider how differences between the interventions may influence their cross-cultural efficacy.

“Furthermore, the interventions differ from each other in important ways, with potential implications for their hypothesized effectiveness across cultural contexts and for our interpretation of the results. For example, the Pluralistic Ignorance and Dynamic Norm interventions harness national-level norms, while the Work-Together intervention references norms among an unspecified referent group. One explanation for why we do not consistently find increased conformity among tighter cultures might be that broader or unspecified referent groups do not elicit a strong conformity response. Indeed, there is ample evidence that people are more likely to adhere to the social norms of relevant and often circumscribed referent groups—such as colleagues for work behaviors, or students for school behaviors—and this tendency may be especially important in tighter cultural contexts [103-105].”

References:

Doell, K. C., Todorova, B., Vlasceanu, M., Bak Coleman, J. B., Pronizius, E., Schumann, P., ... & Lutz, A. E. (2024). The International Climate Psychology Collaboration: Climate change-related data collected from 63 countries. *Scientific data*, 11(1), 1066.

<https://doi.org/10.1038/s41597-024-03865-1>

Vlasceanu, M., Doell, K. C., Bak-Coleman, J. B., Todorova, B., Berkebile-Weinberg, M. M., Grayson, S. J., Patel, Y., Goldwert, D., Pei, Y., Chakroff, A., Pronizius, E., van den Broek, K. L., Vlasceanu, D., Constantino, S., Morais, M. J., Schumann, P., Rathje, S., Fang, K., Aglioti, S. M., Alfano, M., ... Van Bavel, J. J. (2024). Addressing climate change with behavioral science: A global intervention tournament in 63 countries. *Science advances*, 10(6), eadj5778. <https://doi.org/10.1126/sciadv.adj5778>

3. Covariates. While the authors controlled for some individual and group-level variables, further consideration should be given to a broader range of potentially confounding factors. Specifically, the following country-level variables might be important to include as covariates: urbanization, Gini, proportion of Environmental Organizations or pollution levels. These factors may be correlated with both the interventions and individual attitudes towards climate change, potentially confounding the observed relationships.

We originally decided to control for GDP and CO₂ emissions to limit the number of predictors in our models. However, we received a similar suggestion from Reviewer 1 (see Comment #5). We have added in the Supplementary Materials (Tables S23-26) additional models controlling for potential national-level confounders, (i.e., the Gini index, the Human Development Index [HDI], the general Environmental Performance Index [EPI], and the Vulnerability index of the Notre Dame Global Adaptation Initiative). Looking at the correlations between all our national-level variables we find some significant correlations between them (see Table S40), thus we decided to re-run the preregistered models adding only one of these new national-level variables per model (see Tables S23-S26). As we demonstrate in the Supplementary Materials, our results largely replicate our primary analyses. The only difference is in the case of tree planting: in the primary analysis, the interaction between cultural tightness and the Pluralistic Ignorance intervention was significant only in the model without covariates, whereas in the new supplemental analysis it remains significant in the covariate model (see Table S26).

We now discuss these results in the “Robustness Analyses” section of the Results and include this text below for reference.

“Furthermore, when controlling for additional country-level variables (i.e., the Environmental Performance Index [96], the Vulnerability Index of Notre Dame Global Adaptation Initiative Country Index [97], the Gini Index [98], and the Human Development Index [99]; see Tables S23-S26), the results remain largely consistent with the main models reported in Table 2. The only difference concerns tree planting: in the primary analysis, the interaction between cultural tightness and the Pluralistic Ignorance intervention was significant only in the model without covariates, whereas in the supplementary analyses it is significant when we control for additional national-level covariates (see Table S26).”

4. Results. The authors report that "Cultural tightness predicts cross-cultural differences in three of our four primary outcome variables in the control condition." This finding warrants further investigation. Why was this correlation calculated for the control condition? Why wasn't the correlation also examined for the experimental conditions? The authors should report the correlations in all conditions and justify the reason behind this.

We calculated correlations in the control condition to establish a baseline relationship between cultural tightness and each outcome when participants were not exposed to normative information. In the experimental conditions the outcomes were measured after the interventions, so any association between cultural tightness and the outcomes reflects a post-treatment effect. The main models test these relationships through interaction effects (see Results section "*Effect of Norm Interventions by Cultural Tightness*"). Specifically, we look at whether the association between cultural tightness and each outcome differs in magnitude between the intervention and the control conditions. These models (reported in Table 2 and Figure 3) are not simple correlations, as they include multiple covariates. However, these results provide a more comprehensive picture of how cultural tightness relates to the outcomes across interventions.

5. Intervention Effects and Inconsistencies. A critical issue of the study is that none of the three interventions significantly increased belief in climate change relative to the control condition, and the Dynamic Norm intervention was the only one to significantly increase support for mitigation policies compared to control. How does this finding align with previous research on social norm interventions and climate change? Do other studies find similar null effects, or are the results contradictory? A discussion of the consistency (or inconsistency) with the existing literature is essential.

We appreciate the opportunity to clarify how our results align with existing literature. As we note in the Discussion, previous studies have indeed found significant effects of each of the intervention variants we tested on climate attitudes and/or behaviors: Sparkman and Walton for dynamic norms [29], Geiger and Swim for pluralistic ignorance [70], and Howe and colleagues for working together norms [69]. However, as we note, these tests were initially conducted in WEIRD or Global North contexts before being deployed globally [35-36]. In line with this, when we limit our analyses to participants in the U.S., all three interventions significantly affect at least one of our specified outcomes (see Table S22), which is consistent with prior literature. The fact that these effects did not generalize to our global sample suggests that these interventions might be culturally attuned to contexts similar to those in which they were initially developed and tested. Thus, while our null intervention effects in the full sample differ from earlier findings, they do not necessarily contradict them, but instead point to potential cultural boundary conditions that have been underexamined in prior work. We have emphasized this point more clearly in the revised manuscript in the Discussion section "*Why did cultural tightness fail to moderate the efficacy of social norm interventions?*", which highlights that the discrepancy is likely due to differences in cultural fit rather than a failure to replicate per se.

“The interventions tested in the present work have been validated by prior studies, albeit primarily in the Global North (see, for example, Sparkman & Walton [29] for the Dynamic Norm intervention, Howe et al. [69] for the Work-Together Norm, and Geiger & Swim [70] for Pluralistic Ignorance). While the interventions were designed to be broadly applicable, they may not generalize across diverse cultural contexts. Indeed, all of the social norm interventions are effective in shifting at least some of our specified outcomes when limiting our sample to the United States (see Table S22), yet these effects do not generalize to the full cross-cultural sample. This suggests that the interventions may be particularly attuned to the United States and potentially other countries in the Global North context and less well-suited to countries in the Global South. Importantly, however, cultural tightness does not perfectly map onto distinctions between the Global North and South, so while this is a general concern for ManyLabs Megastudies, it does not represent a direct confound for the present study.

Some studies outside of the United States have found that dynamic norms, togetherness norms, and descriptive norm information can be effective means of shifting behaviors (e.g., [28, 101-102]); however, these interventions may need to be culturally-attuned to each setting to be effective—including potentially delivered via different modalities beyond impersonal, light-touch, online surveys. This could suggest that ManyLabs Megastudies, while important for systematically testing interventions across cultural contexts, are not ideally suited to identify locally-optimized and culturally-attuned interventions, especially since many of the tested interventions have been primarily validated in the Global North and proposed by scholars working in those settings. In this study, local samples were mostly collected by local researchers. However, the intervention text was submitted by experts who were largely based in the Global North. To better ensure that interventions are applicable and comparable across cultural contexts, future studies should pre-test and validate intervention materials. Additionally, interventions could be more strategically designed to theorize and test cross-cultural differences in the underlying mechanisms that make some interventions effective in one setting but not in others.”

Furthermore, the interaction effects between cultural tightness and the interventions are inconsistent across different outcome variables. These inconsistent results highlight the complexity of social norm persuasion and the potentially unstable nature of cultural values as moderators. While the authors conducted robustness analyses, these inconsistencies persist. The most pressing challenge may be to address the differences between the interventions and outcomes, and to investigate the underlying mechanisms that might explain these variations. It could also be helpful to determine if the different interventions target different levels of the same variable.

We want to thank the Reviewer for this comment because it made us realize the wording in the results section was unclear in stating that while the associations in each condition differed, we observed null effects consistently in more than three-quarters of all analyses for the predicted moderations. We revised the Discussion section to clarify the overall pattern of

results, and highlight that our few significant effects do not stand up to corrections for familywise error so we see consistently null effects—and based on our Bayesian analyses, these null effects are meaningful.

“Specifically, we find that only one intervention—the Work-Together Norm—was positively moderated by cultural tightness, and for only one of four outcomes: climate change belief. Perhaps most notable, and contrary to our hypotheses, we find that the Pluralistic Ignorance condition has stronger effects on policy support and climate change belief in looser contexts. However, these patterns should be interpreted with caution as these significant interactions are the exception and are not robust to family-wise error corrections. Furthermore, a priori power analyses and a Bayesian analysis suggest that our null effects are not due to lack of statistical power and provide robust evidence of the absence of an effect.”

As noted in our response to the Reviewer’s first point, we have also expanded the Discussion section to consider how specific features of each intervention may interact with cultural context.

6. Alternative model. An intriguing alternative to consider is whether cultural tightness might function more appropriately as an IV, with the different intervention strategies acting as moderators. This approach could provide a different perspective on how cultures independently influence behavior, and how the impacts of these behaviors can be moderated by outside influences.

While we report our results by framing the norm interventions as the independent variable moderated by cultural tightness, one could conceptualize cultural tightness as the independent variable and the interventions as the moderator. Statistically, however, these models are equivalent: in both cases the effects are captured by the interaction between cultural tightness and the intervention condition, with both variables entered as independent variables. The choice of which variable to label the “independent variable” and which the “moderator” is therefore a matter of interpretation and framing rather than a substantive difference in our analysis.

Our primary research question is to ask whether the effectiveness of the three norm interventions depends on cultural tightness, which motivated the framing we adopted in Figure 3. At the same time, we agree that the Reviewer’s proposed perspective is informative for understanding how interventions may amplify or attenuate the effect of cultural context on our outcomes. Reflecting this alternative conceptualization, our existing exploratory analyses on the three outcomes with significant effects (belief, policy support, and tree planting) speak to this interpretation by examining differences among the three interventions within tight and loose cultural contexts (defined as $\pm 1SD$ from the mean). As described in our Results, we find the following effects in cultural contexts with high and low tightness.

“...we find that respondents in the Pluralistic Ignorance condition did not differ from control at high levels of cultural tightness ($p = .63$) and reported greater belief in climate change than the control group in loose cultures ($b = 2.30$, 95% CI [0.67 - 3.94], $SE = 0.84$, $p = .006$).

[...]

As with climate change belief, there are no significant differences in policy support between respondents in the Pluralistic Ignorance and control conditions at high levels of cultural tightness ($p = .47$), though respondents in the Pluralistic Ignorance condition reported significantly higher levels of policy support relative to those in the control condition in culturally loose contexts ($b = 1.85$, 95% CI [0.51 - 3.20], $SE = 0.69$, $p = .006$).

[...]

At high levels of cultural tightness, the Pluralistic Ignorance condition resulted in less tree planting compared to the control condition ($b = -0.26$, 95% CI [-0.48 - -0.03], $SE = 0.11$, $p = .02$), but did not differ from control in culturally loose settings ($b = 0.03$, 95% CI [-0.19 - 0.27], $SE = 0.01$, $p = .75$).

The results of this analysis are displayed in Figure 4, in which we plot the intervention effects separately for high- and low-tightness context for each outcome. These plots allow readers to directly see how cultural tightness impacts each outcome, with the interventions potentially moderating its influence.

Thus, while the underlying statistical model is the same, we present results in both framings, with Figure 3 emphasizing interventions as the primary manipulation, and Figure 4 emphasizing cultural tightness as a contextual factor.

7. Theoretical Contribution. Overall, the researchers should carefully consider the theoretical contribution of their findings to the broader literature on social norm interventions and cultural psychology. What new insights does this study offer for understanding the complex interplay between culture, social influence, and climate change attitudes and behaviors?

We appreciate this prompt to clarify our theoretical contribution. In our newly added Conclusion section, we now include the following sentence to make our contribution to theory more explicit: *“Our findings advance theory by suggesting that conformity to social norm messages is not uniformly amplified in tight relative to looser cultures.”* To help hone this large body of research on culture and norms, we also highlight that cultural influences on conformity to norms may be more likely to be observed elsewhere (e.g., as an individual-level moderator, or as a context-specific moderator):

“Additionally, while it is not possible to experimentally manipulate national levels of cultural tightness, future research might attempt to replicate the theorized interaction using pseudo-experimental designs to examine differences in conformity to norm interventions between people in the same cultural context who come from tight or loose backgrounds [66]. Our null results for national levels of cultural tightness do not imply that individual-level tightness would also yield null results, and testing this possibility directly remains an important next step for future research.”

While prior work suggests that norm intervention effects should be stronger in tighter cultures due to a greater emphasis on norm adherence, our results suggest a potentially more complex and nuanced relationship between cultural tightness and social norm information, that is likely

sensitive to the design and content of the norm intervention. We address this in our new Discussion section “*Why did cultural tightness fail to moderate the efficacy of social norm interventions?*”.